# A Survey of State Representation Learning for Deep Reinforcement Learning

**Ayoub Echchahed**                                                        *ayoub.echchahed@mila.quebec*
*Mila - Québec AI Institute, Université de Montréal*

**Pablo Samuel Castro**                                                              *psc@google.com*
*Mila - Québec AI Institute, Université de Montréal*
*Google DeepMind*

**Reviewed on OpenReview:** *https: // openreview. net/ forum? id= gOk34vUHtz*

## Abstract

Representation learning methods are an important tool for addressing the challenges posed by complex observations spaces in sequential decision making problems. Recently, many methods have used a wide variety of types of approaches for learning meaningful state representations in reinforcement learning, allowing better sample efficiency, generalization, and performance. This survey aims to provide a broad categorization of these methods within a model-free online setting, exploring how they tackle the learning of state representations differently. We categorize the methods into six main classes, detailing their mechanisms, benefits, and limitations. Through this taxonomy, our aim is to enhance the understanding of this field and provide a guide for new researchers. We also discuss techniques for assessing the quality of representations, and detail relevant future directions.

## Contents

## Prologue

### What is State Representation Learning? Why is it useful?

In sequential decision-making systems, state representation learning (SRL) is the process of learning to extract meaningful, task-relevant information from raw observations. In other words, these algorithms aim to distill complex sensory inputs processed by a decision-maker into compact, structured representations, prioritizing critical features while filtering out irrelevant ones. For example, consider a simulated autonomous vehicle navigating a busy urban environment, encountering diverse stimuli like traffic conditions, pedestrians, and changing weather (as shown in Fig. 1).

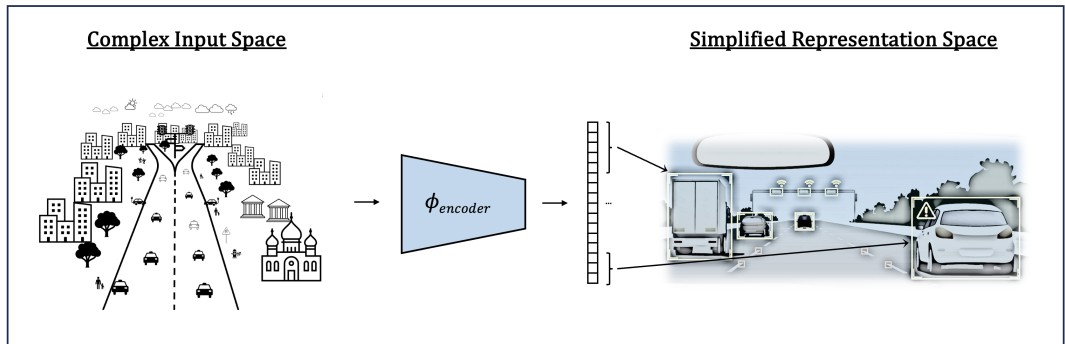

**Figure 1:** Simplified intuition for SRL: Raw sensory inputs from a busy environment are distilled into compressed, task-relevant representations, enabling improved decision-making.

While the color of buildings or the type of roadside trees might be detectable, these details are irrelevant to effective navigation. Instead, elements such as the positions and velocities of vehicles, traffic light statuses, road signs, and pedestrian movements are essential for effective decision-making. SRL could ensure that these crucial variables are emphasized in the learned representation, enabling policies to focus on what matters.

By reducing the complexity of the input space, SRL can enhance learning efficiency, generalization, and robustness to environmental variations (e.g., altered street layouts or weather conditions), enabling autonomous driving agents trained on cities A and B to transfer more easily their learned control policies to cities C and D either zero-shot or with minimal fine-tuning. However, extracting the appropriate features from high-dimensional observations remains a challenging problem, often addressed manually in applications such as robotics and autonomous driving. SRL methods seek to automate this process, producing decision-making systems that are scalable, efficient, and adaptive.

# 1 Introduction

The use of deep reinforcement learning (DRL) for complex control environments has several challenges, including the processing of large high-dimensional observation spaces. This problem, commonly referred to as the "state-space explosion", imposes severe limitations on the efficacy of traditional end-to-end RL approaches, which learn actions or value functions directly from raw sensory inputs, such as pixel observations. As environments grow increasingly complex, these end-to-end methods demonstrate progressively worse data efficiency and generalization, even in response to minor environmental changes. Addressing these limitations is therefore essential for scaling RL to real problems that inherently involve complex and noisy input spaces.

In response to these challenges, recent research has focused on decoupling representation learning from policy learning, treating them as two distinct problems. This strategy has proven effective for managing complex observations by transforming raw inputs into structured representations that retain essential information for decision-making while discarding irrelevant details. By improving data efficiency, state representation learning (SRL) accelerates training, enhances generalization across tasks, and strengthens DRL robustness.

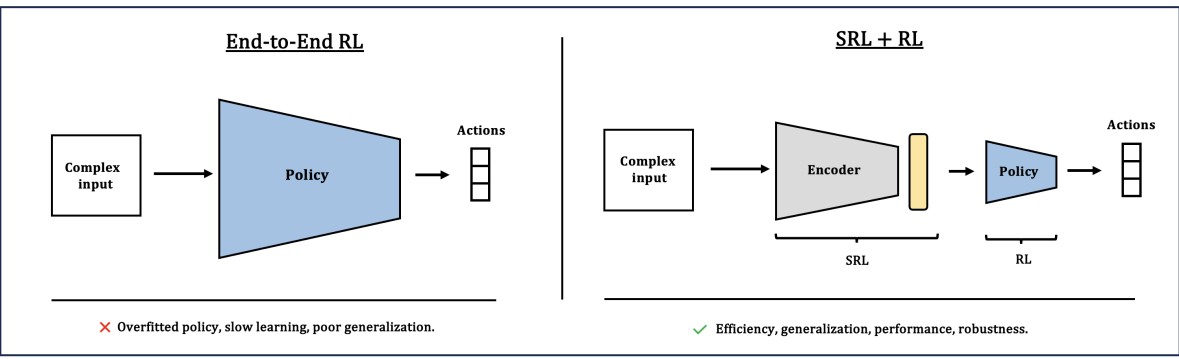

**Figure 2:** Comparison of End-to-End RL (left) and SRL+RL (right). End-to-end directly maps high-dimensional inputs to actions, while SRL separates representation learning and policy learning.

**Motivation.** In recent years, there has been a growth in methods that integrate improved representation learning into DRL, using various approaches. However, many works present inconsistent structuring and categorization of these approaches in their related work sections, making it challenging to obtain a clear and comprehensive understanding of the field. To our knowledge, past surveys provide valuable information but either do not cover the latest developments or focus exclusively on specific SRL classes. Many also group together approaches that rely on fundamentally different principles, blurring key distinctions between them. (Lesort et al., 2018; Ni et al., 2024; Böhmer et al., 2015; de Bruin et al., 2018; Botteghi et al., 2022)

This survey therefore builds on those preceding works by providing an updated and structured analysis of the different approaches in state representation learning for DRL, organizing them based on their principles and effectiveness in different scenarios. Through a detailed taxonomy, we analyze the inner-workings of these classes, highlighting their potential to improve the performance, generalization, and sample efficiency of deep-RL agents. We also explore different ways of evaluating the quality of learned state representations, and discuss promising directions for the field. Overall, this survey can serve as a good resource for researchers and practitioners looking to familiarize themselves with this field.

**Organization.** This work is structured as follows: Section 2 introduces the foundational concepts of state representation learning (SRL) within the deep reinforcement learning (DRL) framework. It defines the problem, objectives, and the characteristics of effective state representations, providing a formal basis for understanding subsequent sections. Section 3 presents the core taxonomy of SRL methods, categorizing them into six primary classes, while elaborating on their mechanisms and highlighting notable work from the literature. Section 4 addresses the critical aspect of evaluation, discussing benchmarks and metrics used to assess the quality and effectiveness of state representations, including their impact on sample efficiency, generalization, and robustness. Lastly, Section 5 explores promising directions for advancing SRL in DRL, such as multi-task learning, leveraging pre-trained visual models, and integrating multi-modal inputs.

## 2 Problem Definition

### 2.1 Formalism

Reinforcement Learning (RL) is typically modeled as a Markov Decision Process (MDP), characterized by the tuple $\langle \mathcal{S}, \mathcal{A}, P, R, \gamma \rangle$. Here, $\mathcal{S}$ denotes the state space, and $\mathcal{A}$ denotes the action space. The transition probability function $P : \mathcal{S} \times \mathcal{A} \to \Delta(\mathcal{S})$, where $\Delta(\mathcal{S})$ denotes the space of distributions over $\mathcal{S}$, defines the probability $P(s'|s,a)$ of transitioning from state $s$ to state $s'$ given action $a$, representing the environment dynamics. The function $R : \mathcal{S} \times \mathcal{A} \to \mathbb{R}$ specifies the immediate reward $R(s,a)$ received after taking action $a$ from state $s$, providing feedback on the action taken.

The objective of an RL agent is to learn a policy $\pi : \mathcal{S} \to \Delta(\mathcal{A})$ that maximizes the expected cumulative discounted reward. Using $\pi$, the agent progressively generates experiences $(s, a, r, s')$, which can be organized into a trajectory $\tau$. For each trajectory $\tau$, the return $G_t$ represents the total accumulated reward from time step $t$ onwards. It is expressed as $G_t = \sum_{k=0}^{\infty} \gamma^k r_{t+k}$, where $\gamma \in [0,1)$ is the discount factor that prioritizes immediate rewards over future ones. To evaluate how good a particular state or state-action pair is, we define value functions. The state value function $V^\pi(s)$ under policy $\pi$ is the expected return starting from state $s$ and following policy $\pi$, given by $V^\pi(s) = \mathbb{E}[G_t|s_t = s]$. Similarly, the action-value function $Q^\pi(s,a)$ represents the expected return starting from state $s$, taking action $a$, and subsequently following policy $\pi$, defined as $Q^\pi(s,a) = \mathbb{E}[G_t|s_t = s, a_t = a]$. Therefore, the objective of the agent can now be expressed as finding an optimal policy $\pi^*$ that maximizes $Q^\pi(s,a)$.

### 2.2 Partial Observability

In many RL settings, full observability is rare. For example, in robotics, sensors might not capture all relevant state factors for optimal decision making in one time-step of data. A POMDP, or partially observable MDP, generalizes the notion of a MDP by accounting for situations where the agent does not have direct access to the full state $s \in \mathcal{S}$ of the environment, hence needing to rely on past observations to infer the current state. Recurrent neural networks (RNNs) are commonly employed to address this partial observability issue, leveraging their hidden state to retain and process information from previous time steps. Another way to handle this is by concatenating the last $n$ observations $(o_t, o_{t-1}, ..., o_{t-n+1})$ to approximate a sufficient statistic for decision-making, thus mitigating the effects of partial observability. For example, agents trained on the ALE benchmark (Bellemare et al., 2013) often employ this technique, known as 'frame stacking'.

In this survey, we adopt a POMDP framework defined as $\mathcal{M} = \langle \mathcal{S}, \mathcal{O}, \mathcal{A}, \mathcal{P}, \mathcal{R}, \gamma \rangle$, where $\mathcal{S}$ is the state space and $\mathcal{O}$ is the observation space. An observation function, $O : \mathcal{S} \to \mathcal{O}$, maps underlying true state to observations, indicating that agents receives only a partial, potentially noisy summary of states. This function establishes the connection between the complete state information and the limited data available to the agent. For an MDP, this function becomes the identity mapping, meaning that each state is fully observable. The transition function $\mathcal{P}$ and the reward function $\mathcal{R}$ are still defined over states, as in an MDP. In this setting, an end-to-end policy $\pi : \mathcal{O} \to \Delta(\mathcal{A})$ maps observations directly to a distribution over actions. This framework is chosen because agents typically do not have full access to the states and must often rely on partial high-dimensional inputs. Other relevant but unaddressed frameworks include Contextual Decision Processes (Krishnamurthy et al., 2016; Jiang et al., 2017) and Block MDPs (Du et al., 2019).

### 2.3 Deep Reinforcement Learning

Deep reinforcement learning (DRL) differs from traditional RL by utilizing deep neural networks to approximate value functions (or policies), either from high-dimensional inputs, or from encoded latent states. This becomes desirable when the state space is large or continuous, which is not well-suited for methods that rely on representing state-action pairs individually in a lookup table, known as tabular reinforcement learning. DRL methods can be broadly categorized into three kind of approaches: value-based, policy-based, and actor-critic methods. Value-based methods, such as Deep Q-Networks (DQN) (Mnih et al., 2013), use a neural network to approximate the action-value function $Q(s,a)$. Policy-based methods, such as REINFORCE (Williams, 1992), directly parameterize the policy $\pi(a|s;\theta)$ and optimize it using gradient ascent on the

expected cumulative reward. Actor-Critic methods combine both value-based and policy-based approaches as they maintain two networks: the actor, which updates a policy $\pi(a|s;\theta)$, and the critic, which evaluates a value function $Q(s,a)$ or $V(s)$. This combination enhances stability and efficiency, making it widely used.

## 2.4 State Representation Learning

The traditional end-to-end approach, which directly maps observations to actions, led to impressive results (Mnih et al., 2013). However, this approach becomes increasingly challenging as the complexity of the environment increases, which is why more efforts are directed towards learning better representations. Representation learning by itself can be defined as the process of automatically discovering features from raw data that are most useful for a task (Bengio et al., 2012). Although representation learning for DRL can be divided into state and action representation learning, the former will be the focus of this survey.

**Problem:** We define the objective of state representation learning (SRL) for reinforcement learning (RL) as learning a representation function $\phi^k : \mathcal{O}_0 \times \mathcal{O}_1 \times \cdots \times \mathcal{O}_k \to \mathcal{X}$, parameterized by $\theta_\phi$, which maps $k$-step observation sequences to a representation space $\mathcal{X}$, thereby allowing us to define policies $\Pi_\mathcal{X}$ over this reduced space. This encoder enables either a policy network $\psi_\pi$ to compute actions $a_t = \psi_\pi(x_t)$, or a value network $\psi_V$ to compute values $v_t = \psi_V(x_t)$, based on the representation $x_t = \phi(o_t)$, instead of directly using high-dimensional observations. The representation $x_t$ is a vector in $\mathbb{R}^d$, where $d$ is the dimensionality of $\mathcal{X}$.

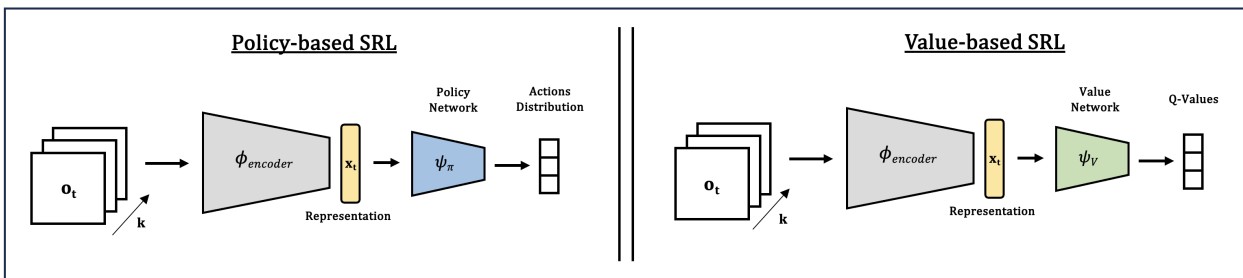

**Figure 3:** Illustration of State Representation Learning (SRL) for RL, where a parametrized transformation $\phi$ is learned, mapping sequences of observations to representations. Two configurations are presented depending if a value-based RL approach is used (right) or a policy-based one (left).

However, not all representation functions are useful to obtain; the goal is to learn an encoder $\phi$ that captures the essential characteristics of effective state representations, as reviewed in the next section. Learning a good encoder simplifies the input space into a compact and relevant representation $x_t$, thereby (1) improving sample efficiency and performance by facilitating the function approximation process performed by the policy/value network; (2) enhancing generalization as learning a policy/value network from representations avoids the overfitting issues seen with high dimensional, unstructured, and noisy observation spaces.

In the presented taxonomy, the focus will be mostly on methods that learn state representations within a model-free online setting, where agents learn representations and policies in real-time through interactions with the environment without using an explicit model of the environment for taking actions. This differs from model-based RL, which involves learning a model of the environment's dynamics that is used for planning, enabling higher sample-efficiency at the cost of higher complexity. Section 5 also explores the offline pre-training of representations. Finally, Table 1 provides a structured overview of key SRL settings.

## 2.5 Defining Optimal Representations

An optimal representation can be defined by its ability to efficiently support policy learning for a set of downstream tasks. The learned space $\mathcal{X}$ should be constrained to a low dimensionality, while remaining sufficiently informative to enable the learning of an optimal policy (or value function) with limited-capacity function approximators. If $\mathcal{X}$ has too much information, it can slow down the learning process and hinder convergence to the optimal policy. Alternatively, a space with insufficient information will prevent convergence to the optimal policy (Abel, 2022). Hence $\mathcal{X}$ ideally balances well information capacity and simplicity.

| Setting | | Description |
|---|---|---|
| Pre-trained | Joint-training | Representations are learned either before RL begins or simultaneously with the RL training objective. |
| Online | Offline | Learning occurs in real-time through interactions with the environment or from pre-collected datasets. |
| Coupled | Decoupled | Encoder parameters are optimized jointly with policy and/or value objectives or independently of them. |
| Reward-based | Reward-free | Representations are influenced by task rewards or focus on environment dynamics and visual features. |
| Single-task | Multi-task | Representations are learned for a specific task or shared across multiple tasks to capture common structures. |
| Model-free | Model-based | Representations are directly used for decision-making or integrated into a world model for planning. |

**Table 1:** Overview of key settings when learning state representations for DRL.

**Structure:** The learned representation space should be structured to encode task-relevant information while remaining invariant to noise and distractions. This means that points within a neighborhood around a representation $x_t$ should exhibit a high degree of task-relevant similarity, which gradually diminishes as the distance from this point increases. These similarities can be encoded in the representation space using information or distances derived from observation features, environment dynamics, rewards, etc. Additionally, the encoder should remain invariant to noise, distractions, or geometric transformations that do not alter the true underlying state of the agent.

**Continuity:** Good latent structure should ensure strong Lipschitz continuity of the value function across nearby representations (Le Lan et al., 2021). In other words, points that are close in the latent space should produce similar value distribution predictions, even if they are distant in the input space. This continuity simplifies the function approximation process performed by the value network $\psi_V$ and promotes better generalization to unseen but nearby states. This similarly applies to the policy network $\psi_\pi$, ensuring that changes in the action distribution occur smoothly within $\mathcal{X}$.

**Sparsity:** Enforcing sparsity constraints on the representations $x_t$ can allow the identification of the most relevant aspects of high-dimensional observations as it encourages the inputs to be well-described by a small subset of features at any given time. This enhances computational efficiency by reducing the number of active features, leading to simpler representations. It also helps avoid overfitting by focusing on the most relevant features, promoting better generalization. Finally, it improves interpretability by making it easier to understand which features drive the decision-making process of agents.

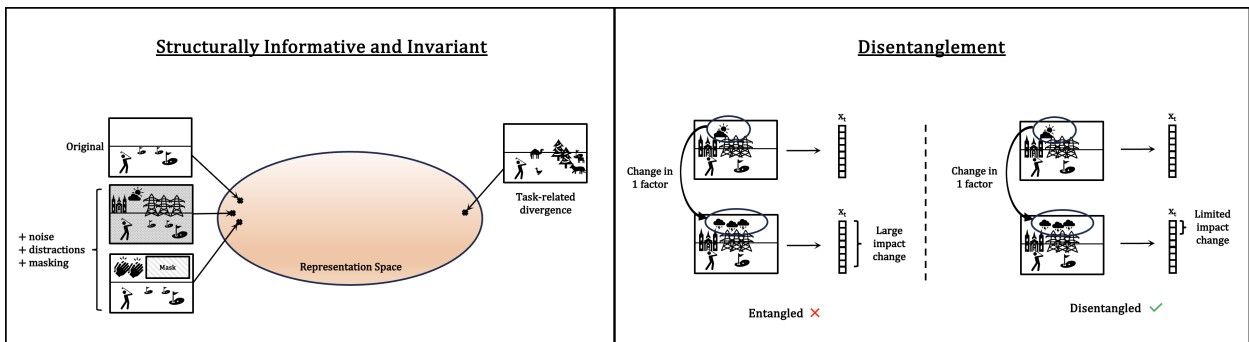

**Figure 4:** Illustration of some optimal state representation properties. The left section demonstrates invariance to noise, distractions, and masking in the representation space, preserving task-relevant information. The right section illustrates disentanglement, where changes in individual factors ideally lead to localized latent impacts.

Previous works have sought to define characteristics of effective representations and abstractions for RL. According to Wang et al. (2024b), optimal representations should exhibit high capacity, efficiency, and robustness. Abel (2022) identifies three essential criteria for state abstractions in RL: efficient decision-making, solution quality preservation, and ease of construction. These criteria stress the importance of balancing compression with performance to facilitate effective learning and planning in complex environments. Other works that discuss characteristics of good representations for RL include (Böhmer et al., 2015), (Lesort et al., 2018), and (Botteghi et al., 2022). Moreover, definitions of optimal representations in the broader context of Self-Supervised Learning (SSL) can often overlap with those needed for control, making them applicable to reinforcement learning as well.

## 3 Taxonomy of Methods

### 3.1 Overview of the Taxonomy

We categorize the representation learning methods into six distinct classes, which are presented in table 2. For each class, we provide a definition, details, benefits, limitations, and some examples of methods. While there are likely other methods in each class, the goal is not to be exhaustive, but rather to focus on the classes themselves. Additionally, some methods may be hybrid, combining techniques from multiple classes.

| Class | Description |
| --- | --- |
| Metric-based | Shape the representation space through a task-relevant distance metric between embeddings. They enhance generalization and efficiency by abstracting states with similar information, reducing complexity. |
| Auxiliary Tasks | Enhance the primary RL task with other simultaneous predictions that indirectly shape representations. These require additional parameters, but can provide accelerated learning on the main task. |
| Augmentation | Leverage data augmentation for learning invariances to geometric and photometric transformations of observations. They do not directly learn representations, but enhance efficiency and generalization. |
| Contrastive | Shape the representation space by learning separate representations for different observations, and similar ones for related inputs. Temporal proximity and/or visual transformations can define similarities. |
| Non-Contrastive | Construct their representation space by only minimizing the distance between the representations of similar observations. Unlike related contrastive approaches, no negative pairs are used during training. |
| Attention-based | Learn attention masks (Bahdanau et al., 2015) for computing scores that highlight important features of the input, helping agents disregard irrelevant details and increase the interpretability of decision-making. |

**Table 2:** Overview of the classes presented in this taxonomy.

### 3.2 Metric-based Methods

**Definition:** Metric-based methods aim to structure the embedding space by using a metric that captures task-relevant similarities between state representations. By mapping functionally equivalent states to similar points in the latent space, these methods can enhance sample efficiency and improve policy learning. For instance, if two different visual observations in a game lead to the same downstream behavior and similar rewards, they can be mapped to the same latent region.

**Details:** The observation encoder, denoted as $\phi_\theta : \mathcal{O} \to \mathbb{R}^n$ with parameters $\theta$, maps observations to an embedding space $\mathcal{X}$ where distances $\hat{d}(\phi_\theta(o_i), \phi_\theta(o_j))$ reflect some task-relevant similarities. For example, the distance metric $\hat{d}$ could correspond to the $L_2$ norm, while the metric could be bisimulation (Ferns et al., 2012), which is introduced below. The representation learning objective can then be formalized as minimizing the expected squared difference between the latent distance $\hat{d}(\phi_\theta(o_i), \phi_\theta(o_j))$ and a metric $d^\pi(x_i, x_j)$ defined over representations (Chen & Pan, 2022), as illustrated in the following equation:

$$L(\phi_\theta) = \mathbb{E}\left[\left(\hat{d}(\phi_\theta(o_i), \phi_\theta(o_j)) - d^\pi(x_i, x_j)\right)^2\right]. \tag{1}$$

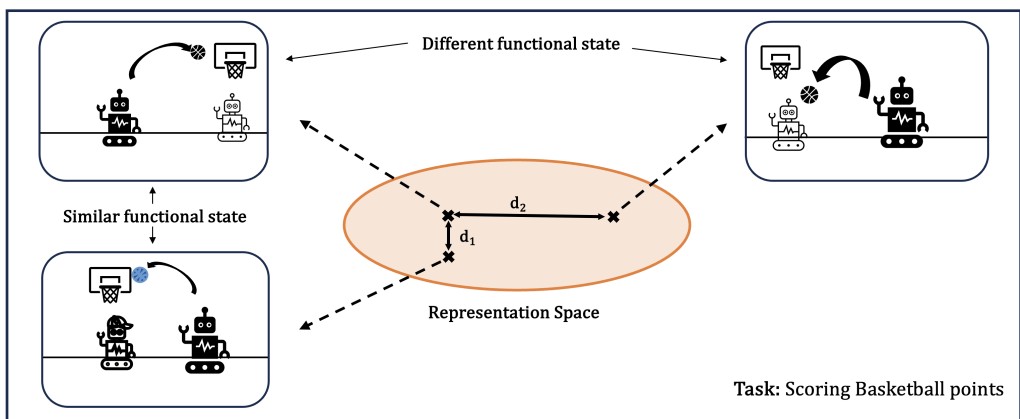

**Figure 5:** Metric-based methods shape the representation space to capture task-relevant information. Representations with similar outcomes (e.g., good basketball trajectory) have minimal distance $d_1$, while those with different outcomes (e.g., wrong basketball trajectory) have a larger distance $d_2$, hence $d_1 \ll d_2$.

**Benefits:** Metric-based methods can offer strong theoretical guarantees by bounding the differences in value function outputs for pairs of embedded states, ensuring that states close in the metric space exhibit similar optimal behaviors. This is formalized as $|V^*(x_i) - V^*(x_j)| \leq d(x_i, x_j)$, which is key to improving sample efficiency and generalization, as it allows the agent to treat behaviorally equivalent states similarly. Additionally, these methods leverage task-relevant MDP information, such as rewards and transition dynamics, to shape the latent state space, making them particularly effective in abstracting away irrelevant visual distractions in more complex environments (Zhang et al., 2021). Additionally, some metric-based methods avoid training extra parameters, providing computational efficiency (Castro et al., 2021).

**Limitations:** The operations involved in certain metrics, such as the Wasserstein distance used in bisimulation, are known to be computationally challenging (Castro, 2020). This can lead to the need for approximations or relaxations, which can weaken the original theoretical guarantees (Chen & Pan, 2022). Furthermore, these methods typically require access to task-specific MDP information, which may not always be readily available or easy to obtain in real-world settings. In fact, even if rewards are available, real-world settings are often characterized by sparse reward structures, which can create latent instability or even embedding collapses in metric-based methods. Embedding explosion is another issue that can affect these methods (Kemertas & Aumentado-Armstrong, 2021). Finally, these methods are impacted by the non-stationary nature of the policy during training, which causes continuous updates to the embedding space and metrics, therefore favoring latent instabilities and hindering consistent performance compared to some other classes.

**Categorization:** Various metrics can be defined to quantify the similarity between states, each influencing how state representations are learned and aggregated.

### a) Bisimulation Metrics

Bisimulation metrics, originally introduced for MDPs by Ferns et al. (2012), offer a way to quantify behavioral similarity between states. By measuring distances between states based on differences in both their rewards

and transition dynamics, it allows state aggregation while preserving crucial information needed for effective policy learning. Formally, the bisimulation metric $d(x_i, x_j)$ between latent states $x_i$ and $x_j$ is updated using the following recursive rule:

$$T_k(d)(x_i, x_j) = \max_{a \in A} \left[ (1-c) \cdot |R(x_i, a) - R(x_j, a)| + c \cdot W_d(P(\cdot|x_i, a), P(\cdot|x_j, a)) \right]. \tag{2}$$

In this formulation, $T_k(d)$ represents an operator that updates the distance function $d(x_i, x_j)$, where $c \in [0, 1]$ is a parameter controlling the balance between the importance of reward differences and transition dynamics. In practice, it is common to set $c = \gamma$, which corresponds to the discount factor in RL, without using $(1-c)$. The term $W_d(P(\cdot|x_i, a), P(\cdot|x_j, a))$ represents the Wasserstein distance (or Kantorovich distance) between the next-state distributions induced by the transitions from states $x_i$ and $x_j$ under action $a$.

Intuitively, $W_d$ can be seen as quantifying the distance between two probability distributions, which corresponds in this case to the next-state distributions of $(x_i, x_j)$. More precisely, the Wasserstein distance is known to measure the cost of transporting one probability distribution to another, and is formalized as finding an optimal coupling between two probability distributions that minimises a notion of transport cost associated with the base metric $d$ (Villani, 2008). By iteratively applying the operator $T_k(d)$, the bisimulation distance $d(x_i, x_j)$ converges to a fixed point $d^*$, yielding the final metric between states that minimizes the loss. This iterative process that progressively shapes the representation space ensures that states with similar rewards and transition dynamics are mapped closer in the representation space, while dissimilar states are mapped further apart. Convergence details and formalism can be found in Castro et al. (2021).

**Methods:** Several methods integrate the bisimulation metric for learning more compact and generalizable representations in reinforcement learning. DBC (Zhang et al., 2021) uses the bisimulation metric to map behaviorally similar states closer in latent space, improving robustness to distractions, but is susceptible to embedding explosions/collapses, and relies on the assumption of Gaussian transitions for metric computation. Kemertas & Aumentado-Armstrong (2021) address these issues by (1) adding a norm constraint to prevent embedding explosion and (2) using intrinsic rewards plus latent space regularization through the learning of an Inverse Dynamics Model (IDM) as an auxiliary task to prevent embedding collapse. The second point is particularly relevant in sparse or near-constant reward settings, where early similar trajectories can incorrectly lead the encoder to assume bisimilarity. A more recent method tackling the sparse-reward challenge in bisimulation-based approaches was introduced by Chen et al. (2024b). Castro et al. (2021) resolves some computational limitations of traditional bisimulation metrics with a scalable, sample-based approach that removes the need for assumptions like Gaussian or deterministic transitions (Zhang et al., 2021) (Castro, 2020), and explicitly learns state similarity without requiring additional network parameters.

**b) Lax Bisimulation Metric**

The lax bisimulation metric (Taylor et al., 2008) extends this concept to state-action equivalence by relaxing the requirement for exact action matching when comparing states, allowing both MDPs to have different action sets, thus providing greater flexibility. For example, Rezaei-Shoshtari et al. (2022) demonstrated the use of this metric for representation learning, which led to improved performance when learning from pixel observations. Le Lan et al. (2021)'s work also highlights why the lax bisimulation metric can provide continuity advantages over the original bisimulation metric.

**c) Related Metrics**

Several alternative metrics have been proposed to shape the representation space of RL agents. For instance, a temporal distance metric was used in Florensa et al. (2019) and Park et al. (2024b), which captures the minimum number of time steps required to transition between states in a goal-conditioned value-based setting. In Rudolph et al. (2024), their action-bisimulation metric replaces the reward-based similarity term of traditional bisimulation with a control-relevant term obtained by training an IDM model, making the approach reward-free. Agarwal et al. (2021a) introduced the Policy Similarity Metric (PSM), which replaces the absolute reward difference in bisimulation with a probability pseudometric between policies and has been shown to improve multi-task generalization.

**d) Impact of distance $\hat{d}$ on Representations**

The choice of how distances between representations are measured often influences the actual nature of the learned representations. For example, the L1 distance, based on absolute differences, promotes sparsity by applying a constant penalty that drives smaller values toward zero, emphasizing distinct features. This can be useful when only a few key features matter in distinguishing states. In contrast, the L2 distance, which uses squared differences, promotes smoother representations by spreading the error across all components, reducing large individual components while retaining contributions from smaller ones. This is more effective when information from all features is relevant, even if some contributions are minor. Some methods instead use orientation-based metrics, such as cosine similarity or angular distance, which can be advantageous in high-dimensional spaces where direction is more significant than magnitude, or where specific properties, such as non-zero self-distances, are desirable (Castro et al., 2021). They can however come with drawbacks, such as embedding norm growth and convergence slowdowns when optimizing cosine similarity, limiting effectiveness without normalization (Draganov et al., 2024). See Table 3 for precise distances formalism.

| Distance $\hat{d}$ | Formula | Description |
|---|---|---|
| L1 (Manhattan) | $\sum_{i=1}^{n} \lvert x_t^{(i)} - x_{t'}^{(i)} \rvert$ | Sum of absolute differences between corresponding vector components. |
| L2 (Euclidean) | $\sqrt{\sum_{i=1}^{n} \left( x_t^{(i)} - x_{t'}^{(i)} \right)^2}$ | Square root of the sum of squared differences between corresponding vector components, emphasizing larger deviations. |
| Angular distance | $\dfrac{\arccos\left( \frac{x_t \cdot x_{t'}}{\lVert x_t \rVert \lVert x_{t'} \rVert} \right)}{\pi}$ | Normalized angle between vectors, ranging from 0 (identical direction) to 1 (opposite direction), capturing rotational differences. |
| Cosine distance | $\dfrac{x_t \cdot x_{t'}}{\lVert x_t \rVert \lVert x_{t'} \rVert}$ | Cosine similarity of vectors, measuring orientation alignment, with values ranging from -1 (opposed) to 1 (aligned). |

**Table 3:** Distance measures $\hat{d}$ used to structure the representation space by quantifying similarities between state embeddings. Each metric defines how distances are measured, but also influences key latent properties.

### 3.3 Auxiliary Tasks Methods

**Definition:** This category is composed of methods that enhance the primary learning task (RL) by having agents simultaneously predict additional environment-related outputs. This is done by splitting the representation part of an agent into $n$ different heads, each with their own set of weights and dedicated task. During training, the errors from these heads are propagated back to the shared encoder $\phi$, guiding the learning in conjunction with the main objective. The role of these predictions is to help agents enrich their representations with additional auxiliary signals.

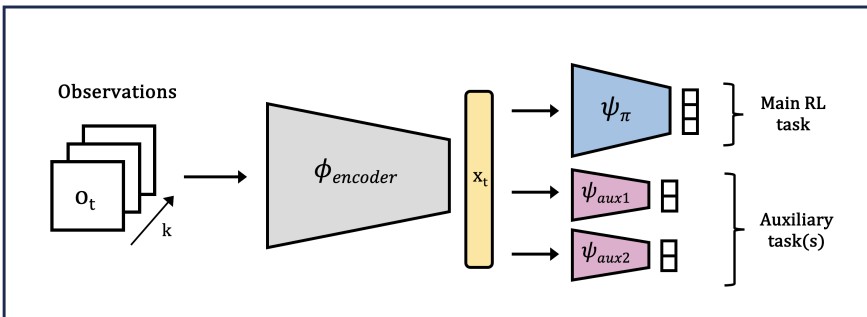

**Figure 6:** The representation $x_t$ of an RL agent is used to make additional predictions on auxiliary task function(s). These predictions are used to improve the representation itself.

**Details:** Let $\mathcal{L}_{\text{primary}}(\theta)$ denote the loss associated with the primary RL objective. Auxiliary tasks are defined as additional functions $\psi_{aux_i}(x_t)$ that, with their own set of parameters $\theta_{Aux} = \{\theta_{\text{aux}_1}, \theta_{\text{aux}_2}, ..., \theta_{\text{aux}_n}\}$, process the representation $x_t$ to output a set of real values with task-dependent dimensions. The loss for each auxiliary task $i$ is represented as $\mathcal{L}_i(\theta_{\text{aux}_i})$, and the overall auxiliary task loss is the sum of all task losses. The combined objective is shown below, with weights $\lambda_i$ balancing primary and auxiliary tasks.

$$\mathcal{L}_{\text{method}}(\theta, \theta_{\text{Aux}}) = \mathcal{L}_{\text{primary}}(\theta) + \sum_{i=1}^{n} \lambda_i \mathcal{L}_i(\theta_{\text{aux}_i}) \tag{3}$$

**Precision:** We define auxiliary tasks as something different than what is called auxiliary losses in the RL literature. Auxiliary losses refers to any loss optimized jointly with the main RL objective, which is something done in methods belonging to most classes here. However, we specifically define auxiliary tasks as additional predictions made during training, using the representation $x_t$ as input, which indirectly enhance the quality of $x_t$. By definition, those supplementary tasks require additional parameters for each task-head, unlike auxiliary losses.

**Benefits:** Auxiliary tasks for RL can enhance the learning process by utilizing additional supervised signals from the same experiences. When faced with environments with sparse rewards, auxiliary tasks can still provide some degree of learning signals for shaping the representation, which increases the learning efficiency of an agent. They can also serve as regularizers, enhancing generalization and reducing overfitting during learning. Finally, they can promote better exploration by guiding the agent toward states that provide more informative signals for the auxiliary tasks.

**Limitations:** However, a downside of using auxiliary tasks to improve representations is the lack of theoretical guarantees when it comes to whether it is actually benefiting the learning process of the main RL objective or not (Du et al., 2020). Defining precisely what makes a good auxiliary task is also in itself a hard problem (Lyle et al., 2021) (Rafiee et al., 2022). Finally, choosing the auxiliary weight(s) that balance(s) the importance of the auxiliary task(s) compared to the main task requires the right tuning of hyper-parameters.

**Categorization:** In the next sections, we explore the inner mechanisms of some class of auxiliary tasks that are commonly employed to learn good state representations in reinforcement learning.

### a) Reconstruction-based Methods

**Definition:** These methods aim to improve state representations by learning to reconstruct original observations $o_t$ using a decoder $\hat{o}_t = \psi_{recon}(x_t)$ that takes as input the encoded representations $x_t = \phi(o_t)$. This reconstruction process can be performed using simple autoencoders (AE), where the objective is to minimize the reconstruction error between the original observation $o_t$ and its predicted reconstruction $\hat{o}_t$. Additionally, it can be achieved using variational autoencoders (VAEs) (Kingma & Welling, 2014), where additional regularization encourages latent variables to follow a predefined distribution (e.g. Gaussian), improving generalization and disentanglement of representations.

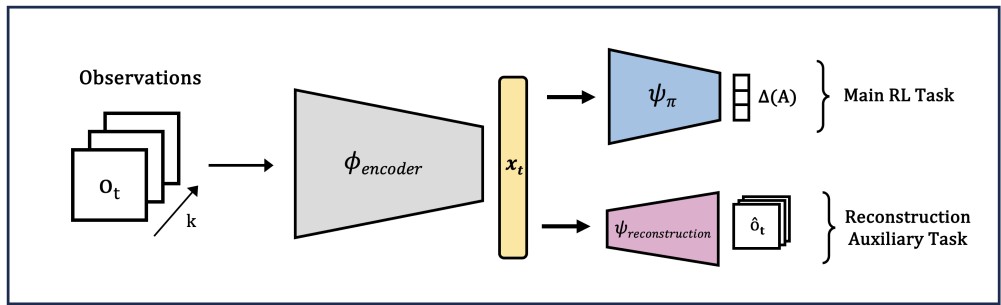

**Figure 7:** Reconstruction as an auxiliary task: The encoder learns compact latent representations by ensuring that the original observation can be reconstructed from the representation.

**Purpose:** These methods enforce the latent space to capture the essential features needed for accurate reconstruction, promoting the learning of compact, denoised representations. This helps the agent to focus on key task-relevant information, improving generalization across environments and enhancing sample efficiency in high-dimensional spaces. Mask-based latent reconstruction avoids the need to reconstruct full observations by focusing the reconstruction only on latent variables, thereby discarding irrelevant features from observational space.

**Failure Cases:** A common failure arises when reconstructing observations that contain significant irrelevant noise or distractions. In such cases, especially when task-relevant features occupy only a small portion of the observation, the model may learn to preserve unnecessary details, leading to poor state representations that degrade learning performance.

**Disentangled Representations:** Reconstruction-based methods can also be used to achieve disentangled representations, where the latent space $\mathcal{X}$ is ideally structured into independent subspaces $\mathcal{X}_i$, each capturing a distinct factor of variation $v_i$ from the observation space. Methods like $\beta$-VAE (Higgins et al., 2016) enforce stronger constraints on the latent space than regular VAEs in order to promote disentanglement, ensuring that changes in one factor (e.g., object color) do not affect others (e.g., arm position), thus enhancing the robustness and adaptability of learned representations in complex environments. More related methods include (Higgins et al., 2017) (Thomas et al., 2018) (Kabra et al., 2021) (Dunion et al., 2023) (Dunion et al., 2024) (Dunion & Albrecht, 2024).

## b) Dynamics Modeling Methods

**Definition:** Dynamics modeling methods use latent forward and inverse models as auxiliary tasks to implicitly improve representations. A latent forward dynamic model (FDM) predicts the next representation $\hat{x}_{t+1} = f(x_t, a_t; \phi_{\text{fwd}})$ from the current representation $x_t$ and action $a_t$, while a latent inverse dynamic model (IDM) predicts the action $\hat{a}_t = g(x_t, x_{t+1}; \phi_{\text{inv}})$ that caused a transition.

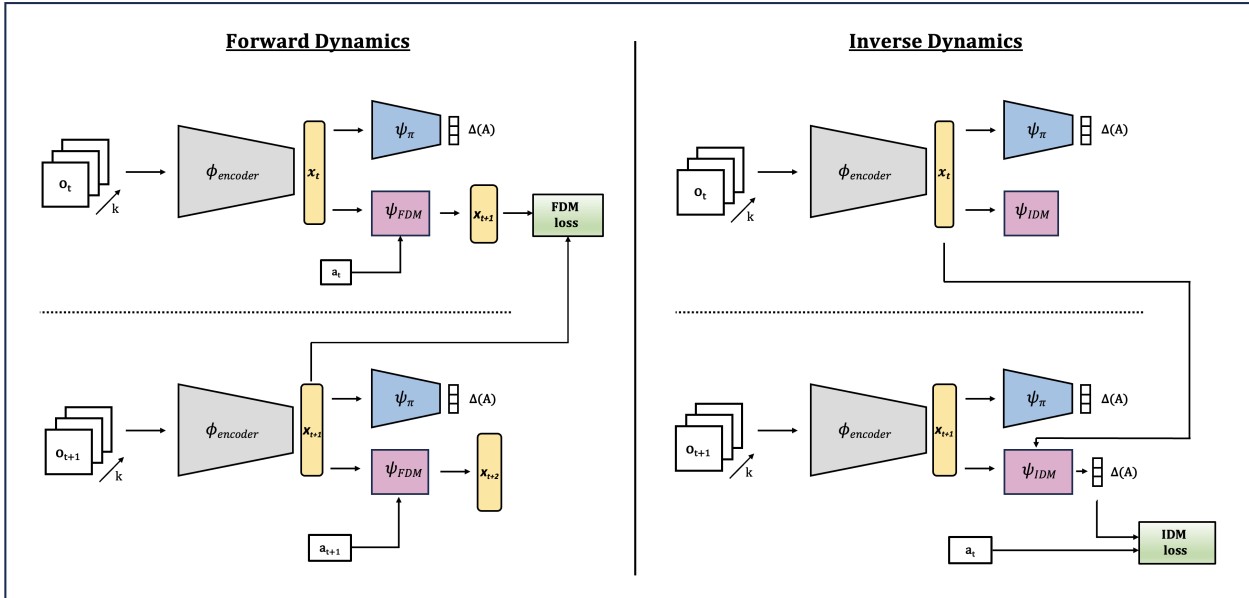

**Figure 8:** Dynamics modeling as an auxiliary task: Forward Dynamics Models predict future representation(s) based on the current representation and action, capturing environment dynamics. Inverse Dynamics Models predict action that caused transitions between representations, emphasizing controllable features.

**Purpose:** FDMs help the agent learn a representation that captures environment dynamics, ensuring that the latent space encodes the essential transition information needed to predict future states. IDMs ensure that the representations encode information to recover the action that led to the state change, focusing on controllable aspects of the environment.

**Failure Case:** A failure case of using FDMs occurs when the transition model lacks a grounding objective, such as reward prediction (Tomar et al., 2021). In such cases, the model can collapse by mapping all observations to the same representation, minimizing the loss trivially, and failing to learn meaningful representations, especially if the critic's signal becomes noisy due to distractions.

**k-step predictions:** Using k-step predictions, where the model predicts multiple future representations instead of just one at each step, can further enhance the representation by capturing longer-term dependencies and improving performance across time (Schwarzer et al., 2020). For IDMs, predicting initial actions from trajectories $o_t$ to $o_{t+k}$ can also ensures positive control properties (Lamb et al., 2023; Islam et al., 2023a).

**Hierarchical Models:** McInroe et al. recently introduced a hierarchical approach utilizing multiple latent forward models (FDMs) to capture environment dynamics at varying temporal scales. Each level in the hierarchy learns a distinct FDM that predicts the representation $x_{t+k}$ k-steps ahead based on previous representations and actions. Additionally, a learned communication module facilitates the sharing of higher-level information with lower-level modules. When compared on a suite of popular control tasks, it achieves noticeable performance and efficiency gains over baseline approaches. Importantly, this differs from predicting all $k$ next representations $x_{t+1}$ to $x_{t+k}$.

### c) More Auxiliary Tasks

A wide variety of additional predictions can be used to support representation learning in RL. Here, we highlight a few additional examples. **(i)** Reward Prediction (Yang et al., 2022) (Zhou et al., 2023) involves predicting the immediate reward $r_t$ based on the current state $x_t$ and action $a_t$, guiding the agent to encode task-relevant features essential for value estimation. This task is especially useful in non-sparse reward settings, where it serves as a discriminator of critical information and benefits from being combined with latent modeling to capture relevant dynamics. **(ii)** Random General Value Functions (GVFs) (Zheng et al., 2021) predict random features of observations based on random actions, generating varied signals that enhance state representations, even when the main RL task is detached through a stop-gradient. **(iii)** Termination Prediction (Kartal et al., 2019) anticipates whether a state will lead to the end of an episode, helping the agent recognize conditions for task completion and improving decisions around critical states. **(iv)** Multi-Horizon Value Prediction (Fedus et al., 2019) involves predicting value functions over multiple future horizons, allowing the agent to account for both short and long-term consequences, supporting more balanced and informed decision-making. **(v)** Proto-Value Networks (Farebrother et al., 2023) help agents learn structured representations by predicting future rewards under random conditions.

### 3.4 Data Augmentation Methods

**Definition:** Data augmentation (DA) methods represent a class of techniques that enhance sample-efficiency and generalization capabilities of RL agents through the manipulation of their observations. By applying geometric and photometric transformations to their inputs, such as rotations, translations, and color shifts, these methods can enforce invariance to irrelevant visual changes.

**Details:** These methods normally introduce an observation transformation function $T$ that generates augmented observations $\tilde{o}$ based on the original observations $o$, where $\tilde{o} = T(o)$. The transformation $T$ is chosen such that it preserves the essential task-relevant properties of $o$. A form of explicit and/or implicit regularization is then used to enforce some degree of Q-invariance and/or $\pi$-invariance. Formally, the invariance of a Q-function with respect to a transformation $f_T$ is defined as $Q(s,a) = Q(f_T(s), a)$ for all $s \in S, a \in A$. Similarly, a policy $\pi$ is considered invariant to $f_T$ if $\pi(a \mid s) = \pi(a \mid f_T(s))$ for all $s \in S, a \in A$.

Two type of strategies can be used to enforce invariance: **(1)** Implicit regularization applies transformations directly to the input data during the training process, using both original and transformed observations to train the network to generalize across these variations (Hu et al., 2024); **(2)** Explicit regularization, on the other hand, achieves invariance by modifying the loss functions to ensure that both the policy (actor) and the value estimates (critic) remain unchanged by the transformations $f_T$. This is done by adding terms in the loss functions that penalize discrepancies between outputs, such as Q-values or action distributions, for both original and transformed inputs.

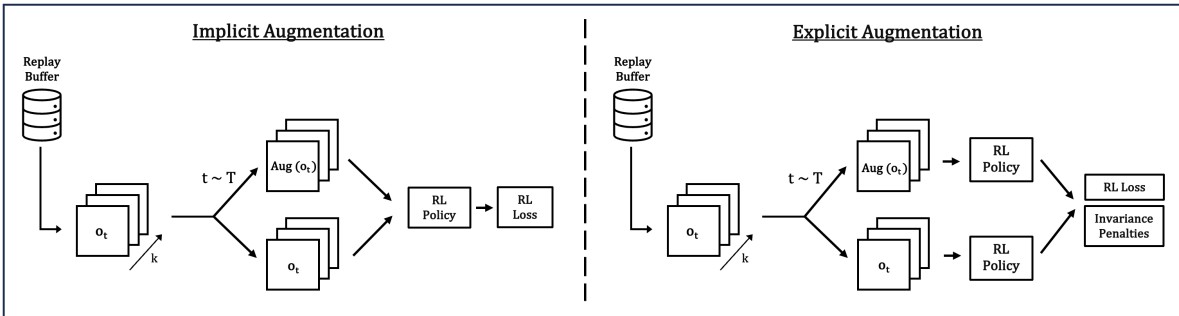

**Figure 9:** Implicit DA (left) augments observations directly used to train the policy and/or value network, promoting robustness through diversity without explicit constraints. Explicit DA (right) augments observations supplemented by regularization penalties that enforce $Q/\pi$ invariances.

**Benefits:** DA-based methods enhance sample efficiency by diversifying training samples, enabling robust policy learning with fewer interactions and reducing overfitting. They improve generalization by simulating visual variations, reducing sensitivity to distribution shifts and aiding adaptation to new settings. Crucially, while remaining simple and effective, the usage of DA was shown to preserve learning plasticity (Ma et al., 2024), an essential aspect with non-stationary objectives.

**Limitations:** These approaches don't directly learn or structure the representation space for incorporating task-specific information, making them fundamentally limited. Additionally, strong augmentations can introduce noise that disrupts training (e.g., high variance in Q-value estimates), and augmentations non-adapted to a task may affect the learning of critical features.

**Augmentations:** Common augmentations applied to observations in DRL include the following: **(i)** Random Cropping, which modifies the image borders without altering central objects; **(ii)** Color Jittering, which adjusts brightness, contrast, and saturation to mimic varying lighting conditions; **(iii)** Random Rotation, involving slight image rotations that do not affect task orientation; and **(iv)** Noise Injection, where stochastic noise is added to images to simulate sensory disturbances or camera imperfections (Ma et al., 2022). Some augmentations, such as random cropping, have shown greater benefits (Laskin et al., 2020), although their effectiveness can be highly task-dependent.

**Methods:** Data-regularized Q (DrQ) (Kostrikov et al., 2020) enhances data efficiency and robustness by integrating image transformations and averaging the Q target and function over multiple transformations, thereby reducing the variance in Q-function estimation. Building on DrQ, DrQ-v2 (Yarats et al., 2021) introduces improvements like switching from SAC to DDPG and incorporating n-step returns, along with more sophisticated image augmentation techniques such as bilinear interpolation to further enhance generalization and computational efficiency. RAD (Laskin et al., 2020), on the other hand, focuses on training with multiple views of the same input through simple augmentations, improving efficiency and generalization without altering the underlying algorithm. DrAC (Raileanu et al., 2021) was introduced as an explicit regularization method that automatically determines suitable augmentations for any RL task and uses regularization terms for both the policy and value functions, enabling DA for actor-critic methods. SVEA (Hansen et al., 2021) proposes an augmentation framework for off-policy RL, which improves the stability of Q-value estimation. Addressing previous limitations, SADA (Almuzairee et al., 2024) enhances stability and generalization by augmenting actor and critic inputs, allowing a broader range of augmentations.

Some methods also rely on more unique techniques: (Li et al., 2024) propose normalization techniques for improved generalization, acting as latent data augmentations by altering feature maps instead of raw observations. To stabilize policy/Q-estimation outputs on augmented observations even further, Yuan et al. (2022a) proposed to identify task-relevant pixels with large Lipschitz constants (by measuring the effect of pixel perturbations on output decisions), and then to augment only the task-irrelevant pixels, which preserve critical information while benefiting from data diversity. Inspired by Fourier analysis in computer vision, Huang et al. (2022) introduced frequency domain augmentations, which provide a task-agnostic plug-and-play alternative to traditional spatial domain data augmentation methods.

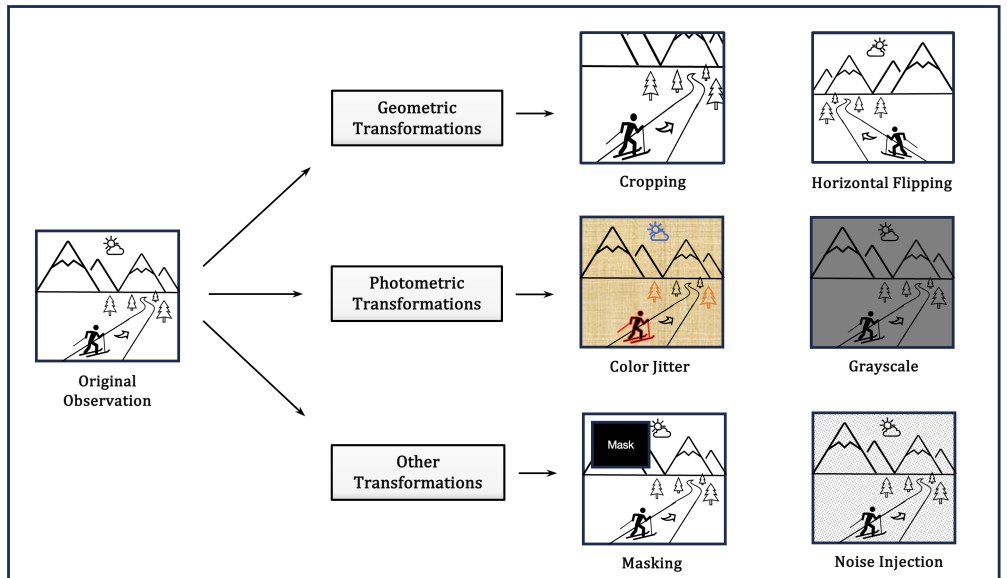

**Figure 10:** Common observation augmentations in RL. Geometric transformations alter spatial properties like cropping or flipping, while photometric transformations alter visual features such as lighting and color. Other augmentations also exist, such as masking or noise injection.

**Precision:** Although DA methods are surveyed here, they do not always rely on an encoder for enhancing efficiency and generalization in RL. This class also focuses on observation augmentations only, although other forms, like transition or trajectory augmentations, can improve learning (Ma et al., 2022; Yu et al., 2021). For more on DA in RL, see Hu et al. (2024) and Ma et al. (2022).

## 3.5 Contrastive Learning Methods

**Definition:** Contrastive learning methods aim to learn effective representations for deep RL agents by contrasting positive pairs (similar data points) against negative pairs (dissimilar data points). This approach utilizes a contrastive loss function that encourages the model to increase the similarity of representations derived from positive pairs while simultaneously decreasing the similarity of representations from negative ones. These methods can leverage different strategies to define positive pairs, such as using data augmentations or exploiting the temporal structures in the data.

**Details:** The InfoNCE loss (van den Oord et al., 2018b) is a widely used contrastive loss for learning representations, both in vision-based SSL and RL specifically. It can be defined as:

$$\mathcal{L}_{\text{NCE}} = -\mathbb{E}_{(o,o^+,\{o_i^-\})} \left[ \log \frac{\exp(\text{sim}(\phi_\theta(o), \phi_\theta(o^+)))}{\exp(\text{sim}(\phi_\theta(o), \phi_\theta(o^+))) + \sum_{i=1}^N \exp(\text{sim}(\phi_\theta(o), \phi_\theta(o_i^-)))} \right]. \tag{4}$$

In the objective above, $o$ represents an observation (anchor), $o^+$ is a positive sample—typically a similar observation to $o$, generated through data augmentations like cropping, rotation, or jittering—and $\{o_i^-\}_{i=1}^N$ are negative samples, which are dissimilar observations selected randomly or based on temporal differences. The encoder $\phi_\theta$ maps observations to representations, while similarity between representations, $\text{sim}(x, x')$, can be assessed using cosine similarity or dot product.

Intuitively, optimizing this loss encourages the model to make the numerator (similarity between $o$ and $o^+$) as large as possible relative to the denominator (which sums the similarities between $o$ and each negative sample). This pushes representations of positive pairs closer together and separates representations of negative pairs, ensuring that observations with similar underlying features cluster in the representation space, while dissimilar observations are spread apart.

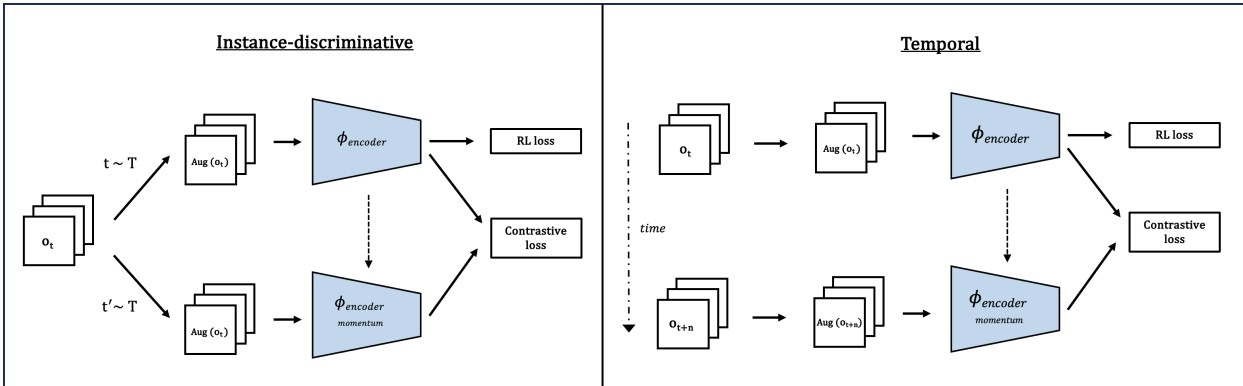

**Figure 11:** Two contrastive learning frameworks: (1) Instance-discriminative contrastive learning with data augmentation (left), and (2) Temporal contrastive learning (right).

**Categorization:** Contrastive methods can be categorized by how they generate positive and negative observation pairs. Instance-discriminative contrastive learning uses data augmentation to create variations of the same observation as positives, with different batch observations as negatives. Temporal contrastive learning leverages the sequential nature of inputs, treating nearby time steps $o_t$ as positives to capture temporal consistency, while distant observations serve as negatives.

**Benefits:** Contrastive methods structure the representation space informatively, either by creating invariance to non-task-relevant variations in observations and/or by making the representation space temporally coherent and smooth. This invariance aspect is especially valuable in complex environments, where different observations might not alter the fundamental true state $s_t$, thus aiding in maintaining consistent decision-making processes.

**Limitations:** Scaling contrastive methods to high-dimensional spaces can be challenging due to the exponential growth of contrastive samples required to learn representations as the input space's dimension grows (LeCun, 2022). Finding appropriate negative pairs can also be a challenge: if negatives are too easy, learning plateaus without gaining useful insights, while overly difficult negatives can hinder learning. Contrastive methods therefore require high batch sizes to avoid biased gradient estimates caused by limited negative samples within a batch (Chen et al., 2022). Finally, because these methods do not leverage reward signals, they may be limited when such are available.

**a) Instance-Discriminative Contrastive Learning**

CURL (Srinivas et al., 2020) is a popular approach that make uses of a contrastive loss to ensure that representations of augmented versions of the same image are closer together than representations of different images, which enforces some beneficial invariance properties in the representation space that improve generalization and efficiency in visual RL tasks. To advance this direction further, future methods could integrate ideas similar to (Wang et al., 2024c), where their notion of augmentation consistency ensures that stronger augmentations push an augmented sample's representation further from the original than weaker ones, structuring the latent space more informatively.

**b) Temporal Contrastive Learning**

CPC (van den Oord et al., 2018a) use autoregressive models to predict future latent states by distinguishing between true and false future states, encouraging representations that capture essential predictive features. Building on CPC, CDPC (Zheng et al., 2024a) introduces a temporal difference estimator for the InfoNCE loss used in CPC, improving efficiency and performances in stochastic environments. ATC (Stooke et al., 2020) aligns temporally close $o_t$ under augmentations, learning representations independently of policy updates, which has proven effective in complex settings.

Additional related approaches, such as ST-DIM (Anand et al., 2019) and DRIML (Mazoure et al., 2020), formulate their objectives based on mutual information maximization between global and local representations (Hjelm et al., 2018). Some methods combine contrastive learning with auxiliary tasks, such as Allen et al. (2021) who combines contrastive learning with an Inverse Dynamics Model (IDM) to learn Markov state abstractions. TACO (Zheng et al., 2024b) takes a different approach and learns both state and action latents by maximizing mutual information between representations of current inputs & action sequences, and the representations of corresponding future inputs.

**Linking Contrastive and Metric-based**

Both metric-based and contrastive methods use some notion of similarity between embeddings to learn $\phi_\theta$, but they define it differently. Contrastive methods use a binary approach, treating pairs of observations as either positive or negative, aiming to minimize representation distances for positives and maximize them for negatives. Metric-based methods, however, quantify similarity more precisely with a continuous distances, often given by a metric that reflects task-relevant information (e.g. rewards, transition). While contrastive methods are task-agnostic, relying on data transformations or temporal proximity, metric-based approaches incorporate task-specific information, enriching $\mathcal{X}$ with task similarity when available.

### 3.6 Non-Contrastive Learning Methods

**Definition:** Non-contrastive methods in RL learn effective representations by minimizing the distance between similar observations, identified through temporal proximity or data transformations, as in contrastive approaches. However, unlike contrastive methods, they do not explicitly push apart dissimilar observations, relying only on positive pairs during training.

**Details:** Methods in this class rely heavily on techniques to prevent total dimensional collapse—a failure mode where the representation space collapses to a single constant vector. This collapse occurs when embeddings are only drawn together using positive pairs, leading to a trivial solution where all embeddings converge to a constant vector, $\phi(o_t) = c$, which minimizes error but retains no meaningful information. To mitigate this, non-contrastive approaches employ two kind of strategies:

**(i)** Regularization techniques, which modify the loss function to preserve embedding diversity, e.g., enforcing a covariance matrix of a batch of embeddings close to identity (Bardes et al., 2021); **(ii)** Architectural techniques, which introduce learning asymmetries via mechanisms such as latent predictors, momentum encoders, and stop-gradients, to regulate learning updates and prevent collapse (Grill et al., 2020).

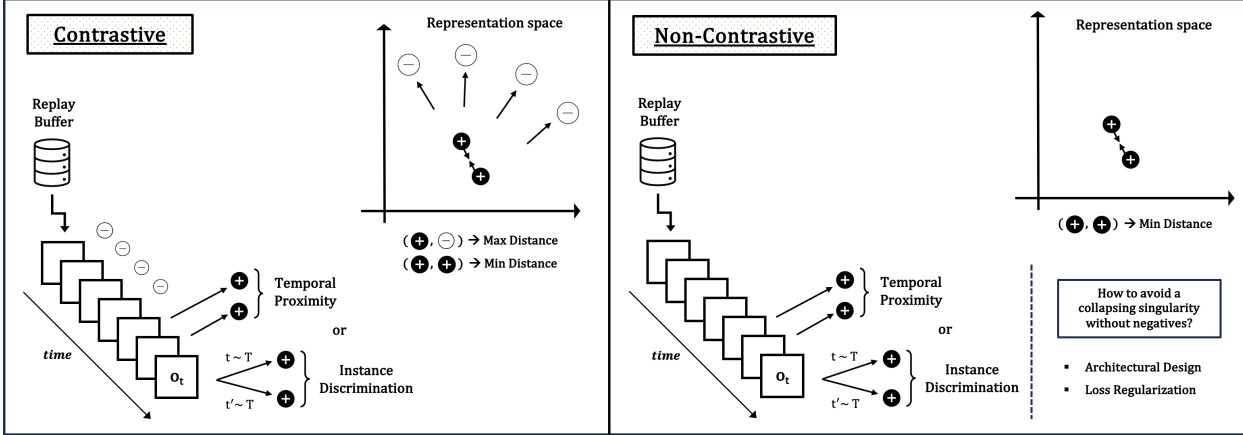

**Figure 12:** Distinction between contrastive and non-contrastive approaches. Contrastive methods rely on both positive and negative pairs to structure the representation space, maximizing similarity within positive pairs and minimizing it for negatives. Non-contrastive methods, which avoid the use of negative pairs, address the challenge of representation collapse through architectural designs or loss regularization. In both approaches, positive samples are generated either through instance discrimination using data augmentations of the same $o_t$ or via temporal proximity.

**Benefits:** Avoiding the need for negative pairs greatly simplifies the learning process, making these methods more computationally efficient and stable in high-dimensional spaces. Additionally, by focusing solely on positive pairs, non-contrastive methods are better suited for scaling to complex observation spaces, avoiding the pitfalls of hard-to-balance negative sampling that can limit contrastive approaches. These approaches are also intuitively aligned with biological representation learning, where positive associations are done without contrasting them with negative examples.

**Limitations:** Non-contrastive methods are susceptible to informational collapse (also known as dimensional collapse), where the embedding vectors fail to span the full representation space, resulting in a lower-dimensional subspace that limits the information encoded. This issue, affecting both contrastive and non-contrastive methods, leads to redundancy in the representation, as embedding components can become highly correlated rather than decorrelated, reducing the diversity and effectiveness of the learned features. Additionally, these methods often lack task-specific information, such as rewards, which could guide the formation of more meaningful state representations.

**Categorization:** Methods in this class can be classified based on whether they make use of a latent predictive component in their architecture or not. In the self-supervised learning terminology, we refer to the former as Joint Embedding Architectures (JEA) and the latter as Joint Embedding Predictive Architectures (JEPA) (Assran et al., 2023). Non-predictive methods aim to make representations invariant to transformations without using a predictor between representation backbones. In contrast, predictive methods incorporate a non-constant predictor, making the representations self-predictive by learning a latent dynamics model during training, which is discarded afterward.

### a) Non-Predictive Methods

BarlowRL (Cagatan & Akgun, 2023) can be seen as an example of a non-contrastive method used for RL that does not use a predictive component. Based on the Barlow Twins framework (Zbontar et al., 2021) and the Data-Efficient Rainbow algorithm (DER) (Hessel et al., 2018), this regularization-based method trains an encoder to map closely together embeddings of an observation $o_t$ and its data-augmented version $o'_t$. Tested on the Atari 100k benchmark, it showed better results than the contrastive framework CURL (Srinivas et al., 2020), but didn't outperform a non-contrastive predictive method presented in the next section called SPR (Schwarzer et al., 2020).

### b) Self-Predictive Methods

Self-predictive methods can be further categorized by whether the predictor relies only on the representation $x_t$ or is also conditioned on transformation parameters between observations, such as the action $a_t$ (Garrido et al., 2024). Without conditioning, methods like BYOL (Grill et al., 2020) and SimSiam (Chen & He, 2021) learn transformation-invariant representations. But conditioning on $a_t$ enables the encoding of action effects on representations by the predictor, capturing the dynamics of the environment. Among approaches, temporal self-predictive methods predict future latent representations by using temporally close observations as positive pairs, encouraging the encoder to capture compressed and predictive information about future states. Specifically, the encoder $\phi(o_t)$ is jointly learned with a latent transition model $P(x_t, a_t)$, which can be extended to predict multiple steps into the future. Augmenting raw observations during training can also enhance positively the robustness of representations and promote richer feature learning.

**Methods:** SPR (Schwarzer et al., 2020), inspired by BYOL (Grill et al., 2020), learns a latent transition model to predict representations from augmented inputs several steps into the future, achieving strong efficiency in pixel-based RL and outperforming expert human scores on several Atari games. PBL (Guo et al., 2020), designed for multi-task generalization, predicts future latent embeddings that recursively predict agent states, creating a bootstrap effect to enhance dynamics learning. SPR and PBL both operate in the latent space, allowing richer and multi-modal learning.

A comprehensive analysis of self-predictive learning in MDPs and POMDPs was conducted by Ni et al. (2024), where they also introduced a minimalist self-predictive approach validated across various control settings, including standard, distracting, and sparse-reward environments. Tang et al. (2023) also explored self-predictive learning in reinforcement learning, specifically highlighting its ability to learn meaningful latent representations by avoiding collapse through careful optimization dynamics. Building on their insights, they introduced bidirectional self-predictive learning, using forward and backward predictions to improve representation robustness. More works in self-predictive representation learning include (Zhang et al., 2024b; Fujimoto et al., 2024; 2025; Khetarpal et al., 2024; Voelcker et al., 2024; Yu et al., 2022).

### 3.7 Attention-based Methods

**Definition:** Attention-based methods in reinforcement learning involve mechanisms that enable agents to focus on relevant parts and features of their complex observations while ignoring less important information. This selective focus allows an agent to process inputs more efficiently, leading to better learning efficiency, performance, and interpretability of an agent's decision making.

**Details:** Attention mechanisms in visual RL agents are typically implemented using mask-based or patch-based attention. Mask-based attention learns weights to highlight relevant regions in observations, while patch-based attention divides inputs into patches and learns relevance scores to focus on the most significant ones. Different types of attention—such as regular or self-attention, soft or hard attention, temporal or spatial attention, single-head or multi-head attention, and top-down or bottom-up attention—can be integrated within an RL agent's architecture. For CNN-based encoders, agents extract feature maps $h_1, h_2, \ldots$ through convolutional layers $c_1, c_2, \ldots$, starting from the observation $o_t$, and map these to the representation $x_t$ using fully connected layers. Self-attention modules can be applied at different stages, targeting low-level or high-level features, or even spanning multiple layers. Alternatively, attention can directly bottleneck inputs (i.e., by learning a mask over raw input patches before any encoding layer to select only the most salient regions).

The computational steps of a self-attention module operating on feature maps (e.g. top of figure 13) are as follows. Starting with a feature map $H$, the module projects $H$ into query, key, and value matrices: $Q = W_q H$, $K = W_k H$, and $V = W_v H$, where $W_q$, $W_k$, and $W_v$ are learned transformations. Attention weights are calculated as $A = \text{softmax}(QK^\top/\sqrt{d})$ where $d$ is the dimensionality of each key vector; $A$ thus reflects the relevance of each region in $H$. The resulting attention-weighted representation $Y = AV$ aggregates information from $V$ according to these relevance scores, focusing on parts of the input that enhance the relevance of $x_t$.

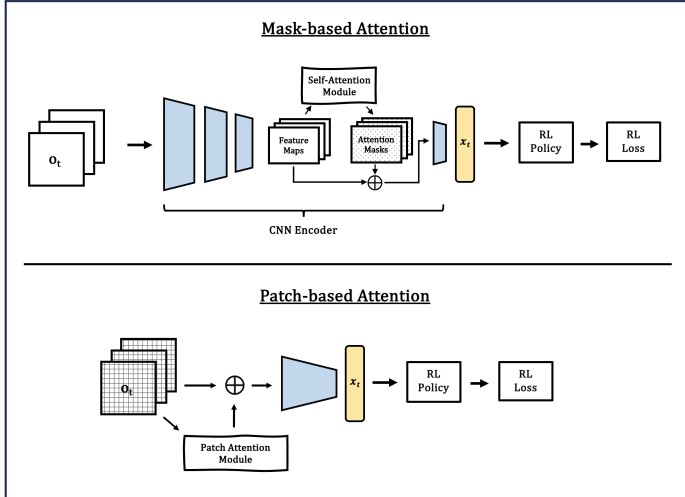

**Figure 13:** Top: A self-attention module operates on high-level feature maps extracted from observations, creating masks that reweight features via element-wise multiplication. Bottom: An attention bottleneck is applied directly to observations, where an attention module selectively focuses on patches of the input.

**Benefits/Limitations:** These methods enhance efficiency by focusing on task-critical regions both temporally and spatially, allowing agents to process relevant observations more effectively and reduce the complexity of the state space. They also improve interpretability, providing insights into the agent's focus area with the usage of saliency maps, making the decision-making process easier to understand. On the other hand, those mechanisms can increase computational complexity due to the additional parameters required. They may also exhibit poor generalization in new settings or with heavy distractions (Tang et al., 2020).

**Methods:** Tang et al. (2020) introduced a self-attention bottleneck that learns to select the top K image patches for efficient processing of relevant visual information. Wu et al. (2021) proposed an attention module that uses a self-supervised approach to generate attention masks, enhancing CNN-based RL performance on Atari games. Chen et al. (2019) integrated temporal and spatial attention into a hierarchical DRL framework for improved lane changing in autonomous driving, achieving smoother and safer behaviors. Chen et al. (2024a) propose Focus-Then-Decide (FTD), a method that combines two auxiliary tasks with the RL loss to train an attention mechanism. Their mechanism selects task-relevant objects from outputs of a foundational segmentation model, leveraging prior knowledge to enhance performance in complex visual scenarios.

Bertoin et al. (2022) propose Saliency-Guided Q-networks (SGQN) for improved generalization, a framework that generate saliency maps highlighting the most relevant parts of an image for decision-making. The training procedure is supported by two additional objectives: a self-supervised feedback loop where the agent learns to predict its own saliency maps and a regularization term that ensures the value function depends more on the identified important regions. Sodhani et al. (2021b) introduced an attention-based multi-task learning method that uses a mixture of k encoders, with task context from a pre-trained language model determining soft-attention weights for combining encoder outputs, improving task performance. Mott et al. (2019) proposed a soft, top-down attention mechanism that enhances interpretability and performance by generating task-focused attention maps, enabling better generalization and adaptability to unseen game configurations, surpassing bottom-up approaches.

### 3.8 Alternative Approaches

This section provides a brief overview of two additional classes of methods for state representation learning in DRL. While less widely common than other classes, they still offer unique insights.

**a) Spectral-based Methods**

**Definition:** Spectral-based methods for state representation learning employ the eigenvalues and eigenvectors of matrices derived from transition dynamics to capture structural and geometric information about the environment. These methods create embeddings that preserve the connectivity and global topology of the state space, enhancing the representation $x_t$ of the observation $o_t$.

**Details:** In these methods, observations $o_t$ can be represented as nodes in a graph $G = (O, W)$, where $W$ is a matrix that reflects transition probabilities between observations. The Laplacian matrix $L = D - W$, where $D$ is a matrix capturing how connected each observation is, provides spectral properties used to create embeddings. By using the smallest eigenvectors of $L$, each observation $o_t$ is mapped to a vector $x_t = [e_1(o_t), e_2(o_t), \ldots, e_d(o_t)]$ that captures both local and global relationships in the environment. Laplacian representations were originally fixed after pretraining but recently enable online adaptation to policies.

**Methods:** Gomez et al. (2024) introduces a framework that approximate Laplacian eigenvectors and eigenvalues effectively, while addressing challenges such as hyperparameter sensitivity and scalability. Wang et al. (2023) improves traditional Laplacian representations by making sure the Euclidean representation distance between two observations also reflects a measure of reachability between them in the environment, allowing better reward shaping in goal-reaching tasks. Wu et al. (2019) propose a scalable and sample-efficient approach for computing Laplacian eigenvectors, enabling practical applications in high-dimensional or continuous environments. Ren et al. (2023) improves spectral methods by learning state-action features that do not depend on the data collection policy, allowing a better generalization across policies. Finally, Shribak et al. (2024) improves spectral representations by using nonlinear diffusion-based approximations, ensuring sufficient expressiveness for any policy's value function.

**b) Information Bottleneck Approaches**

**Definition:** The Information-Bottleneck (IB) principle (Tishby et al., 2000) provides a framework for learning compact and task-relevant representations by optimizing the trade-off between compression and relevance. When used for SRL, IB aims to learn a state encoder that minimizes the mutual information $I(O; X)$ between observations $O$ and representations $X$ to compress irrelevant information while maximizing $I(X; Y)$, where $Y$ corresponds to task-relevant targets.

**Details:** The IB objective balances compression and relevance of state representations by minimizing $I(O; X) - \beta I(X; Y)$, where $o_t$ and $x_t$ denote observations and representations, respectively. Regular IB requires estimating mutual information terms, which is computationally intractable for high-dimensional inputs. Variational Information Bottleneck (VIB) (Alemi et al., 2017) addresses this by introducing parametric approximations with an encoder $q_\phi(X|O)$ and decoder $p_\psi(Y|X)$. The VIB objective therefore combines task relevance and compactness, making it scalable for DRL.

**Methods:** REPDIB (Islam et al., 2023b) leverages the IB principle by incorporating discrete latent representations to enforce a structured and compact representation space. It maximizes the relevance of task-specific information while filtering out exogenous noise, leading to improved exploration and performances in continuous control tasks. DRIBO (Fan & Li, 2022) employs a multi-view framework to filter out irrelevant information via a contrastive Multi-View IB (MIB) objective, enhancing robustness to visual distractions. IBAC (Igl et al., 2019) integrates IB into an actor-critic framework, promoting compressed representations and better feature extraction in low-data regimes. Additional methods include IBORM (Jin et al., 2021), which leverage IB in a multi-agent setting, and MIB (You & Liu, 2024), which introduced a multimodal IB approach for learning joint representations from egocentric images and proprioception data.

## 3.9 Comparison and Practical Guidelines

The table below provides a brief glimpse of practical guidelines for selecting an SRL method that belongs to the main classes of methods presented before. Metric-based methods are most effective when dense rewards and known dynamics are available, offering provable abstraction guarantees but suffering under sparse or non-stationary rewards. Contrastive methods deliver strong invariances when large batches of positive and negative pairs can be formed, but depend on careful negative mining and high batch sizes. Auxiliary-task methods supply additional learning signals in sparse-reward or exploration-intensive settings, at the cost of extra hyperparameter tuning. Data augmentation methods improve robustness in visual domains with distractors but do not directly shape the latent space. Non-contrastive methods avoid the need for negative samples yet require architectural or loss-based measures to prevent collapse. Attention-based methods focus computation on task-relevant regions and enhance interpretability, though they introduce additional parameters and may overfit.

| Class | Use Case | Pros | Cons |
|---|---|---|---|
| Metric-based | Known dynamics | Provable generalization | Sparse-reward failure |
| Auxiliary Tasks | Sparse rewards | Extra learning signal | Hyperparameter tuning |
| Data Augmentation | Visual distractors | Simple & robust | No latent shaping |
| Contrastive | Ample negatives | Strong invariance | Negative mining |
| Non-Contrastive | Low-batch regimes | Low memory footprint | Collapse risk |
| Attention-based | Selective focus | Interpretability | Parameter overhead |

**Table 4:** Comparison of SRL Classes with Use Cases, Benefits and Trade-Offs.

# 4 Benchmarking & Evaluation

Evaluating correctly state representation learning methods requires appropriate benchmarks and tools to assess the quality of the learned representations. Key properties like reward density, task horizon, or the presence of distractions constitute important aspects to consider when choosing benchmarks. Ideally, comparisons between methods should primarily focus on environment-specific properties, as suggested by Tomar et al. (2021). Therefore, instead of claiming universal superiority on a benchmark, it is better to state that experiments show an approach is better suited for learning under visual distractions (e.g., noisy backgrounds).

The next sections review common SRL evaluation aspects and methods for assessing representation quality.

## 4.1 Common Evaluation Aspects

Evaluating state representation learning methods requires assessing their effectiveness in achieving key objectives during and after RL training. Comparisons typically focus on the following aspects:

**Performance:** In DRL, better performance is defined by achieving a higher expected cumulative reward. An improved representation $\phi$ should enable a policy $\pi_\phi$ that maximizes reward, such that $J(\pi_\phi) \geq J(\pi_{\text{baseline}})$. This applies whether the representations are pre-trained via SRL and kept fixed during RL, or when performance is evaluated as the representations are learned.

**Sample Efficiency:** Data efficiency can be quantified by the number of samples $N$ required to achieve a specific performance level. Let $N(\epsilon)$ be the number of samples needed to achieve a performance within $\epsilon$ of the optimal performance $J^*$. An improved representation $\phi$ enhances sample efficiency if $N_\phi(\epsilon) < N_{\text{baseline}}(\epsilon)$, meaning fewer samples are needed with the improved representation to achieve the same performance.

**Generalization:** The generalization ability of a representation quantifies its capacity to support policy performance in previously unseen environments. A representation generalizes well if the policy $\pi_\phi$ achieves a consistent expected return across different environments, measured by the condition $|J(\pi_\phi; \text{train}) - J(\pi_\phi; \text{test})| \leq \delta$, where $\delta$ is a small tolerance threshold. Additionally, fine-tuning with minimal environment interactions to achieve prior high performance further indicates the effectiveness of a representation in this context. Generalization may also be assessed by metrics such as transfer success rate or zero-shot adaptation score.

**Robustness:** The robustness of a state representation $\phi$ can be assessed by its stability across variations in underlying RL algorithms, hyperparameters, and training conditions within the SRL method. Formally, given a set of configurations $C$ (e.g., different RL algorithms, hyperparameter settings, or noise levels) and a tolerance $\delta$, the representation is considered robust if $\max_{c \in C} |J(\pi_\phi; c) - J(\pi_\phi; c^*)| \leq \delta$, where $c^*$ represents an optimal configuration. A robust representation exhibits minimal performance variance across $C$, indicating reduced dependency on specific settings and greater applicability across various RL scenarios.

## 4.2 Assessing the Quality of Representations

Evaluating the quality of learned state representations in reinforcement learning (RL) is essential for understanding how well representations capture task-relevant information. Therefore, having good metrics for quantifying the quality of those learned state representations is crucial. Various methods can be utilized for this purpose. We categorized those based on whether they require access to ground truth states $s_t$ or not, which is not always a realistic assumption to make.

### a) Evaluation Without True States

**Total Return:** The most common approach for evaluating the quality of learned state representations is simply to let an RL agent use the learned states to perform the desired task and assess the final return obtained with specific representations methods. This verifies that the necessary information to solve the task is embedded in the representation $x_t$. However, this process is often time-intensive and computationally demanding, necessitating substantial data and multiple random seeds to account for the high variance in performance (Agarwal et al., 2021b). The performance can also vary depending on the base agent, adding further complexity to this evaluation approach.

**Visual Inspection:** Another technique that does not require access to ground truth states involves extracting the observations of the nearest neighbors in the learned state space and visually examining if those observations match approximately to the close neighbors in the ground truth state space (Sermanet et al., 2018). In other words, close points in the latent state space should correspond to observations with similar task-relevant information.

**Latent Information:** More general metrics for assessing the quality of representations can be based on measuring properties of good SSL representations, such as the variance of individual dimensions across a batch, the covariance between representation dimensions, the average entropy of representation vectors, or spectral properties of a representation matrix, such as its effective rank or condition number (Garrido et al., 2023). Disentanglement could be measured by perturbing randomly small parts of an input observation and measuring the impact on the dimensions of representation $x_t$, as disentangled representations are expected to limit the effect of random small perturbations to only a few dimensions, reflecting independent and meaningful feature encoding.

**Latent Continuity:** Metrics can also be designed to evaluate the continuity and smoothness of Q-functions, action predictions, or simply temporal coherence in the representation space (Le Lan et al., 2021). By examining multiple local areas in the representation space, along with the nearest neighbors and their corresponding Q-values, action distributions, or time-steps, such metrics can assess whether nearby points yield similar values or actions. Ensuring this continuity helps maintain stable decision-making and simplifies the function approximation of the policy $\psi_\pi(x_t)$ and/or value network $\psi_v(x_t)$, enhancing efficiency.

**Linear Probing:** Zhang et al. (2024a) proposed to use 2 probing tasks for assessing the quality of learned representations: (i) reward prediction in a given state, and (ii) action prediction taken by an expert in a given state. The authors used linear probing specifically, where a linear layer is trained on top of frozen representations for each prediction task, constraining the probe's performance to rely heavily on the quality of $x_t$. Overall, their probing tasks were shown to strongly correlate with downstream control performance, offering an efficient method for assessing the quality of state representations. Probing on frozen representations $x_t$ can also assess latent information by reconstructing observations. However, this technique may be less effective with noisy or distracting inputs.

**Interpretability:** Understanding the key information that RL agents focus on within observations and encoded representations can provide better insights into their decision-making. A general scheme for determining the attention levels at different parts of an observation consist of perturbing random areas of the input, then measuring the resulting policy or value changes (Greydanus et al., 2018) (Yuan et al., 2022a). Regions causing higher variance under similar perturbations are likely more relevant for the agent. This perturbation-based approach can also be extended to individual representation dimensions to evaluate their importance by analyzing induced changes in policy/value outputs. With stacked observations, gradient-based techniques (Weitkamp et al., 2019) can offer a practical alternative via action-specific activation maps.

## b) Evaluation with True States

**Probing:** Evaluating the quality of learned state representations can be done by training a linear classifier on top of frozen representations to predict ground-truth state variables (Jonschkowski et al., 2017), and reporting metrics such as the mean F1 score. This linear probing approach was applied by Anand et al. (2019) to evaluate the performance of a representation learning technique in Atari. The underlying assumption is that successful regression indicates that meaningful features are well-encoded in the learned state, and good generalization performance on the test set suggests a robust encoding of these features. This concept can also be extended to non-linear probes (Tupper & Neshatian, 2020).

**Geometry:** The KNN-MSE (K-Nearest Neighbors Mean Squared Error) metric from Lesort et al. (2017b) can evaluate learned state representations by first identifying the k-nearest neighbors of each image $I$ in the learned state space. It then calculates the mean squared error between the ground truth states of the original image $I$ and its nearest neighbors $I'$ to assess the preservation of local structure in the learned representations. Manifold learning metrics such as NIEQA (Zhang et al., 2012) can also evaluate how well the learned representations preserve the original state's local and global geometry (Lesort et al., 2017a).

**Disentanglement:** The metric from Higgins et al. (2016) evaluates how well a representation separates factors of variation by fixing one factor in data pairs, measuring representation differences, and predicting the fixed factor with a linear classifier. Higher accuracy indicates better disentanglement.

## 5    Looking Beyond

While current state representation learning (SRL) methods for deep reinforcement learning (DRL) have made good progress in improving sample efficiency, generalization, and performance, there is still room for improvement. As environments become more complex and varied, it is important to explore more ways of enhancing those techniques for more challenging settings. This section looks at several directions, each extending the learning of state representations to broader domains.

| Direction | Description |
|---|---|
| **Multi-Task** | Explore the sharing of representations across multiple tasks to capture common structures. |
| **Offline Pre-Training** | Leverage datasets of past interactions for pre-training state representations, boosting efficiency and transfer. |
| **Pre-trained Vision** | Integrate representations from pre-trained visual models into agents for efficiency and generalization gains. |
| **Zero-Shot RL** | Produce representations that enable agents to perform new tasks without additional training. |
| **Leveraging Priors** | Utilize large language models (LLMs/VLMs) to incorporate prior knowledge into representations. |
| **Multi-Modal** | Methods that integrate information from multiple sensory modalities for getting richer representations. |

**Table 4:** Promising directions for enhancing state representation learning in DRL.

### 5.1    Multi-Task Representation Learning

**Definition:** Multi-task representation learning (MTRL) involves training an RL agent to extract a shared low-dimensional representation among a set of related tasks and use either one or separate heads attached to this common representation to solve each task. This approach leverages the similarities and shared features among tasks to improve overall learning efficiency and performance. Although various settings of MTRL exist, they often share the common points presented here.

**Benefits:** MTRL reduces sample complexity by exploiting shared task structures, enhancing learning convergence, generalization, robustness to new tasks, and enabling knowledge transfer, where learning one task improves performance on related ones (Cheng et al., 2022) (Efroni et al., 2022).

**Challenges:** Negative transfer when shared representations are suboptimal for certain tasks represents an important issue, which can lead to interference and degraded performance (Sodhani et al., 2021a). Balancing task contributions to the shared encoder is also challenging when tasks vary in difficulty or nature. Additionally, differences in data distributions among tasks can limit the effectiveness of representations, with benefits often relying on some assumptions (Lu et al., 2022).

**Methods:** To mitigate negative interference, CARE (Sodhani et al., 2021a) proposes to encode observations into multiple representations using a mixture of encoders, allowing the agent to dynamically attend to relevant representations based on context. Efroni et al. (2022) introduced a framework for efficient representational transfer in reinforcement learning, showcasing sample complexity gains. Kalashnikov et al. (2022) presented MT-Opt, a scalable multi-task robotic learning system leveraging shared experiences and representations. PBL (Guo et al., 2020) trains representations by predicting latent embeddings of future observations, which are simultaneously trained to predict the original representations, enabling strong performances in multitask

and partially observable RL settings. Hessel et al. (2019) introduced PopArt, a framework that automatically adjusts the contribution of each task to the learning dynamics of multi-task RL agents, hence becoming invariant to different reward densities and magnitude across tasks. Ishfaq et al. (2024) proposed MORL, an offline multitask representation learning algorithm that enhances sample efficiency in downstream tasks. Cheng et al. (2022) introduced REFUEL, a representation learning algorithm for multitask RL under low-rank MDPs, with proven sample complexity benefits.

## 5.2  Offline Pre-Training of Representations

**Definition:** The offline pre-training of state representations refers to the learning of state representations from static datasets of trajectories $\{(o_{i,t}, a_{i,t}, r_{i,t}) \mid i = 1, \ldots, N; t = 1, \ldots, T\}$ or demonstrations $\{(o_{i,t}, a_{i,t}) \mid i = 1, \ldots, N; t = 1, \ldots, T\}$ in order to accelerate learning on downstream RL tasks. This strategy is motivated by the necessity to enhance data efficiency on downstream tasks and overcome the limitations of learning tabula rasa, which often leads to some degree of overfitting. Akin to human decision-making, this aims to leverage prior knowledge contained in some already collected interactions.

**Benefits:** By leveraging large amounts of pre-collected data, offline pre-training of representations can enhance data efficiency by reducing the need for extensive online interactions to achieve high performance. This pretraining process can lead to better initializations for RL algorithms, resulting in faster convergence and superior final performance on downstream tasks. Moreover, representations learned from diverse offline datasets can enhance RL agents' robustness, improving generalization across environments and tasks.

**Challenges:** The quality and diversity of the offline dataset are crucial, as poorly curated or biased datasets can result in suboptimal representations. Ensuring that the learned representations are transferable and useful for a wide range of downstream tasks is also complex, as certain features may not generalize well beyond the pretraining context. Additionally, the pretraining of large models on extensive datasets demands substantial computational resources, making the process both time-consuming and expensive.

**Methods:** The study performed by Yang & Nachum (2021) demonstrate that offline experience datasets can successfully be used to learn state representations of observations such that learning policies from these pre-trained representations improves performance on a downstream task. Through their investigation, they demonstrate performance gains across 3 downstream applications: online RL, imitation learning, and offline RL. Their results provide good insights on different representation learning objectives, and also suggests that the optimal objective depends on the downstream task's nature and is not absolute. Kim et al. (2024) also investigated the efficacy of various pre-training objectives on trajectory and observation datasets, but focused specifically on evaluating the generalization capabilities of visual RL agents compared to a broader range of pre-training approaches. Farebrother et al. (2023) introduced Proto-Value Networks (PVNs), a method that scales representation learning by using auxiliary predictions based on the successor measure to capture the structure of the environment, producing rich state features that enable competitive performance with fewer interactions. Schwarzer et al. (2021) introduced SGI, a self-supervised method for representation learning that combines the latent dynamics modeling from SPR (Schwarzer et al., 2020), the unsupervised goal-conditioned RL from HER (Andrychowicz et al., 2017), and inverse dynamics modeling for capturing environment dynamics. It achieves strong performance on the Atari 100k benchmark with reduced data, and good scaling properties.

## 5.3  Pre-trained Visual Representations

**Definition:** Pre-trained visual representations (PVRs), also called visual foundation models for control, involve utilizing unlabeled pre-training data from images and/or videos to learn representations that can be used for downstream reinforcement learning tasks. These representations are trained to learn the spatial characteristics of observations $\{o_i \mid i = 1, \ldots, N\}$ and the temporal dynamics from videos $\{o_{i,t} \mid i = 1, \ldots, N; t = 1, \ldots, T\}$, and can be seen as initializing an agent with some initial vision capabilities before learning a task. PVRs can be pre-trained on domain-similar data or general data with transferable features.

**Benefits:** PVRs benefit from abundant and inexpensive image and video data compared to action-reward-labeled trajectory data, enabling scalable learning across domains. They improve sample efficiency by pro-

viding pre-learned visual features, reducing task-specific relearning. PVRs also enhance generalization by transferring robust visual features across environments, even under variations or unseen conditions.

**Challenges:** Downsides include the lack of temporal data leveraged in Image-based PVRs, while video-based PVRs often struggle with exogenous noise (e.g., background movements) that degrade performance. Without action labels, distinguishing relevant states from noise becomes significantly harder, and sample complexity for video data can grow exponentially (Misra et al., 2024). Additionally, distribution shifts between pre-training and target tasks further complicate video representation learning (Zhao et al., 2022). Finally, while PVRs benefit model-free RL, Schneider et al. (2024) found they fail to enhance sample efficiency or generalization in model-based RL, especially for out-of-distribution cases.

**Methods:** Yuan et al. (2022b) demonstrated that frozen ImageNet ResNet representations combined with DrQ-v2 (Yarats et al., 2021) as a base algorithm can significantly improve generalization in challenging settings, though fine-tuning degraded performance. MVP (Xiao et al., 2022) showed that pre-training on diverse image and video data using masked image modeling (He et al., 2022) while keeping the weights of the visual encoder frozen preserves the quality of representations and accelerates RL training on downstream motor tasks. Majumdar et al. (2023) studied PVRs across tasks, finding: (1) no universal PVR method dominate despite overall better performance than learning from scratch; (2) scaling model size and data diversity improves average performance but not consistently across tasks; (3) adapting PVRs to downstream tasks provides the best results and is more effective than training representations from scratch. The study made by Kim et al. (2024) of different pre-training objectives suggest that image and video-based PVRs improve generalization across different target tasks, while reward-specific pre-training benefits similar domains but performs poorly in different target environments. Some methods are also exclusively designed for video-based PVRs. Misra et al. (2024) analyzed different approaches for video-based representation learning and found that forward modeling and temporal contrastive learning objectives can efficiently capture latent states under independent and identically distributed (iid) noise, but struggle with exogenous noise, which increases sample complexity. Their empirical results confirm strong performance in noise-free settings but a degradation under noise. Other relevant work includes VIP (Ma et al., 2023) and R3M (Nair et al., 2022), though the latter was initially limited to behavior cloning. Finally, approaches that learn latent representations while recovering latent-action information solely from video dynamics (Schmidt & Jiang, 2024) represent an interesting avenue for video-based PVRs trained on large action-free dataset.

### 5.4 Representations for Zero-Shot RL

**Definition:** Zero-shot RL aims to enable agents to perform optimally on any reward function provided at test time without additional training, planning, or fine-tuning. The objective is to train agents that can understand and execute any task description immediately by utilizing a compact environment representation derived from reward-free transitions $(s_t, a_t, s_{t+1})$. When a reward function is specified, the agent should use the learned representations to generate an effective policy with minimal computation. More precisely, given a reward-free MDP $(S, A, P, \gamma)$, the goal is to obtain a learnable representation $E$ such that, once a reward function $r : S \times A \to \mathbb{R}$ is provided, we can compute without planning, from $E$ and $r$, a policy $\pi$ whose performance is close to optimal.

**Benefits/Challenges:** Zero-shot RL offers flexibility by enabling agents to adapt to various tasks without retraining, enhancing efficiency and scalability across numerous tasks without additional training or planning. However, challenges include developing comprehensive representations without reward information, ensuring transferability across complex tasks, consistently achieving near-optimal performances, or assuming access to good exploration policies during pre-training.

**Methods:** Forward-Backward (FB) representations, introduced by Touati & Ollivier (2021), enable zero-shot RL by learning two functions: a forward function to capture future transitions and a backward function to encode paths to states, trained in an unsupervised manner on state transitions without rewards. The intuition for these representations can be seen as aligning the future of a state with the past of another by maximizing $F(s)^\top B(s')$ for states $s$ and $s'$ that are closely connected through the environment's dynamics. This approach offers a simpler alternative to world models, enabling efficient computation of near-optimal

policies for any reward function without additional training or planning, though it relies on an effective exploration strategy.

In a related study by Touati et al. (2023), FB representations have shown to deliver superior performances across a wider range of settings compared to methods based on successor features (SFs), which also aim to do zero-shot RL based on successor representations (SRs) (Dayan, 1993). Recently, Jeen et al. (2024) proposed a Value-Conservative version of FB representations, addressing the performance degradation issue of previous methods when trained on small, low-diversity datasets. Other similar recent works include Proto Successor Measure (Agarwal et al., 2024), Hilbert Foundation Policies (Park et al., 2024a), and Function Encoder (Ingebrand et al., 2024).

### 5.5 Leveraging Language Models' Prior Knowledge

**Definition:** Leveraging Vision-Language Models (VLMs) and Large Language Models (LLMs) for state representation learning involves using large pre-trained models to transform visual observations into natural language descriptions. These descriptions serve as interpretable and semantically rich state representations, which can then be used to produce task-relevant text embeddings, allowing reinforcement learning (RL) agents to learn policies from embeddings rather than pixels.

**Benefits/Challenges:** Leveraging VLMs/LLMs for this purpose can improve generalization by creating invariant representations that are less affected by disturbances in observations. Indeed, these models can successfully extract the presence of objects and filter out irrelevant details based on task-specific knowledge, focusing only on what is relevant. This has the advantage of leveraging the vast prior knowledge embedded in those large models, and can also enhance interpretability by allowing for a more transparent understanding of the agent's decision-making. However, many challenges still exist for a successful integration of VLMs/LLMs within an RL framework, starting by the higher computational resources necessary to leverage the capabilities of those models.

**Methods:** An interesting work in this direction was done by Rahman & Xue (2024) where they proposed to use a VLM to generate an image description, refined by a LLM to remove irrelevant information, and used for producing a state embedding given to the reinforcement learning agent. Their method showed generalization improvements compared to an end-to-end RL baseline. Other related works include (Chen et al., 2024c; Wang et al., 2024a).

### 5.6 Multi-Modal Representation Learning

**Definition:** Multi-Modal State Representation Learning (MMSRL) integrates multiple data types to create richer and more comprehensive state representations for RL agents. By combining diverse information sources with different properties, MMSRL can enhance an agent's understanding of the environment, improving decision-making and generalization. For example, a robot navigating a room with a camera and a microphone will be able to learn unified representations combining sight and sound with MMSRL. Hence if it hears glass shatter but doesn't see it, the robot will be able to infer danger in another room.

**Benefits/Challenges:** MMSRL creates richer representations by capturing more comprehensive environmental features, making it resilient to noise or occlusion in individual modalities. This robustness enhances generalization and adaptation, improving performance in complex environments.

**Methods:** The work done by Becker et al. (2024) introduces a framework that enables the selection of the most suitable loss for each modality, such as using a reconstruction losses for low-dimensional proprioception data and a contrastive one for images with distractions.

### 5.7 Other Directions

Several directions fall outside the scope of this work but still deserve consideration: (**i**) Exploration strategies for enhanced state representation learning, rather than for rewards directly, will be essential for future open-ended applications, as they can ensure a relevant state space coverage and mitigate the risk of learning good representations for only a small part of the state space. In fact, the interplay between effective explo-

ration and high-quality state representations is particularly important since effective exploration relies on a solid understanding of previously encountered states. (**ii**) State representation learning in continual learning settings, where representations are learned from continually evolving environments, aligns more closely with the dynamic nature of real-world problems and should be investigated further. (**iii**) Evaluating the scalability of SRL approaches remains an open challenge, with future methods needing to scale with increasing environment complexity, number of tasks, and aspects like computational resources, data, and parameters.

Moreover, action spaces, like observation spaces, can become high-dimensional in complex environments. Learning compact action representations has been shown to improve generalization over large action sets (Chandak et al., 2019) by allowing agents to infer the effects of novel actions from those of similar actions encountered earlier. An important future direction is to jointly learn state and action representations, producing dual informative latent spaces that capture both environmental dynamics and action semantics to further enhance policy efficiency and transfer.

## 6 Conclusion

This survey provides a comprehensive overview of techniques used for representation learning in deep reinforcement learning, focusing on strategies to enhance learning efficiency, performance, and generalization across high-dimensional observation spaces. By categorizing methods into distinct classes, we have highlighted each approach's mechanisms, strengths, and limitations, clarifying the landscape of SRL methods and serving as a practical guide for selecting suitable techniques. We also explored evaluation strategies for assessing the quality of learned representations, especially as techniques are applied to increasingly challenging settings. Robust evaluation remains essential for real-world applications, supporting reliable decision-making and generalization.

Looking forward, SRL methods must adapt to a broader set of settings, such as those outlined in Section 5. For each direction in that section, we reviewed related work that could serve as a foundation for further exploration, emphasizing the importance of continued research in these areas. Ultimately, advancing SRL will be crucial for developing robust, generalizable, and efficient DRL systems capable of tackling complex real-world tasks. We hope this survey serves as a resource for researchers and practitioners aiming to deepen their understanding of SRL techniques and offers a strong foundation for learning representations in DRL.

**Limitations:** This survey primarily examines state representation learning methods within the model-free online RL setting, without addressing model-based approaches and offline RL evaluation. The comparisons between classes are also largely theoretical or rely on previous studies. Future work could include experimental evaluations to compare approaches on multiple aspects. Lastly, while the taxonomy provides an overview of the main classes for learning state representations in RL, it does not explore each class in detail, as each could be the focus of its own survey. Some isolated approaches may also be missing due to our focus on categorizing mostly the recent developments.

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
