# OpenReview forum: "A Survey of State Representation Learning for Deep Reinforcement Learning"
_TMLR — Accepted by TMLR_

### Review · Reviewer_sRW8 · 2025-04-29

**Summary Of Contributions:**

This survey paper provides a comprehensive overview of representation learning methods for model-free reinforcement learning. The paper starts off by formulating the reinforcement learning problem. It then dives into a taxonomy of representation learning methods for RL, broadly categorizing the methods into six categories: metric-based, auxiliary tasks, augmentation, contrastive, non-contrastive, and attention-based. For each category, the paper provides an overview of its definition, benefits and limitations, as well as a summary of prominent methods. In the next section, the paper reviews benchmarks and metrics for assessing the quality of the representations, ranging from task performance to similarity metrics. The following section summarizes the emerging directions of RL representations, such as pretrained visual representations and representations for zero-shot RL. Overall, the survey covers a broad array of contemporary methods in a methodological manner, providing a comprehensive guide for researchers in the field.

**Audience:**

Yes

**Broader Impact Concerns:**

The paper surveys relevant representation learning methods for model-free RL. While RL has many real-world implications, this paper's contribution is non-technical in nature and does not require a broader impact statement.

**Claims And Evidence:**

Yes

**Requested Changes:**

- The paper can be strengthened by adding a section on when to use each representation learning method / what methods perform well for a specific use case. This will provide a more practical guide for researchers in the field.

**Strengths And Weaknesses:**

**Strengths**
- The paper covers a wide range of contemporary methods and evaluation techniques for representation learning in model-free RL. The topics are fairly comprehensive, far exceeding my knowledge of the field. I find the categorization particularly helpful for gaining a systematic understanding of the field.
- The paper is well written and easy to read.

**Weaknesses**
- The paper does not go into much details for the specific methods in each category. It's more like a glossary than an instruction manual. It would help to have a section providing intuition for when to use each representation learning method / what methods perform well for a specific use case.

---

### Review · Reviewer_G9du · 2025-05-12

**Summary Of Contributions:**

This paper surveys recent state representation learning techniques (SRL) in the context of deep reinforcement learning (DRL), more specifically in the model-free setting. The paper covers the problem definition, the existing methods, the evaluations, and what's next for this research field. The authors propose a taxonomy for the methods found in the literature. They also make use of several figures to support and explain the different topics discussed in the paper. Overall, this survey paper aims at depicting the recent research in this field as accurately as possible and provides several future directions one can act upon to extend it.

As a researcher in an adjacent field, it is my impression that I got a great deal of information from this survey paper. I'm certain other in the TMLR community will get something out of reading it as well. I recommend this paper for acceptance.

**Audience:**

Yes

**Broader Impact Concerns:**

I don't think there is any concern on ethical implications of this work, i.e. a survey paper on existing methods.

**Claims And Evidence:**

Yes

**Requested Changes:**

## Typos
- p.10: "an IDM as an auxiliary" but IDM is only defined at p.18
- p.12: (Kingma & Welling, 2022) -> That paper was accepted at ICLR 2014.
- fig.9: "D.A" -> DA
- p.15: "camera imperfections Ma et al. (2022)" -> "camera imperfections (Ma et al., 2022)"
- fig.10: "such as cropping" -> "such as masking"?
- p.16: "SSL" is never defined.
- p.20: "softmax(QK^T|\sqrt{d})", should mention what $d$ stands for here.
- p.21: "focus area with the usage saliency maps" -> "...  usage of saliency maps"?
- p.26: "from SPR Schwarzer et al. (2020)" -> "from SPR (Schwarzer et al., 2020)"
- p.26: "Misra et al. (2024)" -> "(Misra et al., 2024)"
- p.26: "Zhao et al. (2022)" -> "(Zhao et al., 2022)"
- p.27: "Agarwal et al. (2024)" -> "(Agarwal et al., 2024)"
- p.27: "Other related works include (Chen et al, 2024c) (Wang et al., 2024a)." -> "... (Chen et al, 2024c; Wang et al., 2024a).

**Strengths And Weaknesses:**

### Strengths

- The paper is well written and well structured paper. The authors goes beyond simply listing the recent works, each section contains benefits/limitations takeaways which is exactly what I look for when reading survey papers.
- I appreciated the many figures illustrating the different techniques and concepts throughout the paper. That helped me validate if my understanding of the text was indeed correct or not.
- As someone that already has some knowledge in state representation learning, I feel I still got valuable insights and knowledge on that topic. To me, this survey is a good starting point to start digging deeper.

### Weaknesses
- Minor. The author clearly stated the focus of the survey was on the state representation learning of the observation space only. Though, I think state representation learning for the action space is going to play an big role in the future, e.g. AI agent using tools or function calls. There might be something to be said about this in the Looking Beyond section. Anyway food for thoughts.

- Minor. For survey papers (especially for journal papers with less restrictions on pages count), it is better to spell out the acronyms of the methods mentioned as reference. That way readers can have a better idea (sometimes) of what each method does. Some acronyms are already spelled out but there are many that are not, it could be great to uniformize this.

- On page 20, in the Attention-based Methods/Details subsection, what does it mean "attention can directly bottleneck inputs"?

- On page 22, in the Benchmarking & Evaluation section, what is "distraction-based learning"? That term does not seem to be mentioned before.

---

### Review · Reviewer_7dyv · 2025-05-14

**Summary Of Contributions:**

The authors lay out the different classes of representation learning techniques in the context of RL. As someone who works with the techniques in RL, I found this work enjoyable to read.

**Audience:**

Yes

**Broader Impact Concerns:**

None.

**Claims And Evidence:**

Yes

**Requested Changes:**

None.

**Strengths And Weaknesses:**

As a survey paper, the work covers different representation learning techniques. To my knowledge, I did not see any parts that were missing.

I do not have any particular weaknesses to point out.

---

### Decision · Action_Editor_8RC1 · 2025-06-07

**Recommendation:** Accept as is

**Additional Comments:**

I (and the reviewers) believe this is a solid survey which deserves to be recognized as such.

**Audience:**

Yes

**Audience Explanation:**

RL is a topic of interest to many in the TMLR community, but is also receiving the renewed attention of the broader ML community thanks to its use in LLMs.

**Claims And Evidence:**

Yes

**Claims Explanation:**

This survey paper offers a comprehensive overview of Deep RL methods related to representation learning.